# Cluster Randomized Designs for One-Sided Bipartite Experiments

**Jennifer Brennan**[*]
Paul G. Allen School of
Computer Science & Engineering,
University of Washington
Seattle, WA 98195
jrbrennan@google.com

**Vahab Mirrokni**
Google Research
New York, NY 10011
mirrokni@google.com

**Jean Pouget-Abadie**
Google Research
New York, NY 10011
jeanpa@google.com

## Abstract

The conclusions of randomized controlled trials may be biased when the outcome of one unit depends on the treatment status of other units, a problem known as *interference*. In this work, we study interference in the setting of one-sided bipartite experiments in which the experimental units—where treatments are randomized and outcomes are measured—do not interact directly. Instead, their interactions are mediated through their connections to *interference units* on the other side of the graph. Examples of this type of interference are common in marketplaces and two-sided platforms. The *cluster-randomized design* is a popular method to mitigate interference when the graph is known, but it has not been well-studied in the one-sided bipartite experiment setting. In this work, we formalize a natural model for interference in one-sided bipartite experiments using the exposure mapping framework. We first exhibit settings under which existing cluster-randomized designs fail to properly mitigate interference under this model. We then show that minimizing the bias of the difference-in-means estimator under our model results in a balanced partitioning clustering objective with a natural interpretation. We further prove that our design is minimax optimal over the class of linear potential outcomes models with bounded interference. We conclude by providing theoretical and experimental evidence of the robustness of our design to a variety of interference graphs and potential outcomes models.

## 1 Introduction

Interference is a well-studied phenomenon in causal inference, whereby the treatment status of one unit can affect the outcome of another. Formally a violation of the Stable Unit Treatment Value Assumption [1], interference has been studied in many settings, including agricultural studies [2], clinical trials [3], social networks [chap. 16 of 4, 5, 6, 7], and marketplaces [8, 9, 10, 11, 12]. Marketplaces exhibit unique forms of interference because they involve two types of units: *buyers* and *sellers*. Units of the same type do not interact directly, but rather their interactions are mediated through their interactions with units of the opposite type. For example, when buyers compete to buy limited goods, an increase in the price one buyer is willing to pay for a good will affect the market-clearing rate for that good, thus increasing the price for all other buyers.

Marketplace experiments can be conceptualized as running an experiment on a bipartite graph between buyers and sellers, with edges representing buyer-seller interactions. Other settings beyond marketplaces can also be formalized as bipartite graphs, such as content platforms (matching viewers to creators) or ride-sharing apps (matching riders to drivers) [13]. We consider perhaps the most

---

[*]Work done while the author was an intern at Google Research, New York, NY.

36th Conference on Neural Information Processing Systems (NeurIPS 2022).

straightforward way of running an experiment on a bipartite graph: by treating and measuring a single side of the graph, a setting we call the *one-sided bipartite experiment framework*. We refer to the units on this side of the graph as *experimental units* since they receive treatment and their outcomes appear in our estimators. We refer to the units on the other side of the graph as *interference units*: while they do not explicitly receive treatment, they mediate the interactions between experimental units.

The presence of interference units distinguishes one-sided bipartite experiments from other settings of network interference because the interactions between experimental units are known in the latter case, but must be inferred from the interactions with interference units in the former. In this work, we show that a popular class of experimental designs for network interference, *cluster-randomized designs*, does not extend immediately to this setting. Instead, we propose a variant of the cluster-randomized design, with a clustering objective that accounts for the way in which interference units mediate the interference between experimental units. We motivate the use of the difference-in-means estimator in this setting and show that clustering according to our objective minimizes the bias of that estimator in a minimax sense over the class of linear potential outcomes models with bounded interference. We further illustrate the failure of existing clustering designs when applied to one-sided bipartite experiments, and conclude by illustrating the robustness of our design to a variety of potential outcomes models and bipartite graphs both theoretically and empirically.

## 1.1 Related work

Accurate estimation of the treatment effect requires some knowledge of the mechanism of interference [14]; otherwise, having even a single treated (resp. control) unit in a graph of control (resp. treated) units could change the outcome of every unit arbitrarily. Existing work varies in the strength and nature of the assumptions on the interference model, but in general these assumptions take one of two types. The *exposure mapping* approach [3, 15], formalized by Aronow and Samii [16], defines a notion of when a unit is "completely treated" or "completely controlled," then uses the inverse propensity score (IPS, also known as Horvitz-Thompson) estimator to construct an unbiased estimate of the average treatment effect. By contrast, an alternate approach is to propose a model for the effect of interference on the potential outcomes, and then rely on this model to estimate the average treatment effect using data from all units (even the ones experiencing a great deal of interference) [6, 17, 18, 19]. Since the quality of the estimate depends on the accuracy of the modeling assumptions, several methods have been developed to estimate the magnitude and form of interference [5, 20, 21, 22, 23, 24]. Chin [25] lessens the reliance on the form of the interference model by developing a model-agnostic regression estimator that reduces average treatment effect estimation to a problem of feature engineering; however, it assumes that each unit's response follows a shared (but unknown) model, which is more restrictive than the potential outcomes model we use in this work.

Even in cases when the model is only approximate, so that including data from units experiencing interference may bias the estimator, Eckles et al. [17] use realistic graph models to argue that the bias incurred is more than offset by the reduction in variance achieved by avoiding an IPS estimator. Our work assumes this same regime, in which the graphs are sufficiently dense that an IPS estimator will be too high-variance to be practical, so we accept an estimator with some bias. We assume a linear model of the potential outcome on the measured exposure, as done in [6, 19]. To avoid strong dependence on the linear assumption, we use the difference-in-means estimator which does not rely on the model of interference, and we choose our experimental design to be minimax optimal over the class of linear potential outcomes models.

Given some model of interference and a choice of estimator, the next question is how to design an experiment (an assignment of units to treatment or control). The most popular experimental design in the case of network interference is the cluster-randomized design, studied by [26, 17, 27] in the case of non-bipartite graphs. This design first clusters units according to the provided graph, and then assigns each cluster to either treatment or control. In the specific case of bipartite graphs, other works propose modifications to unit-level randomization by choosing which side of the graph to randomize [28, 10, 29]. The authors of [18, 19] look directly at clustered designs on bipartite graphs, but they study a two-sided experimental framework in which one side of the graph is randomized while the other is measured. Perhaps closest to our work is that of Rolnick et al. [9], which suggests a balanced partitioning of geographical regions using a clustering objective that is similar to ours. Their work

considers a more restrictive form of the potential outcomes model, and uses clustering heuristics that are specific to the geographical setting. By contrast, our work considers the robustness of the design to a broader class of potential outcomes models and extends beyond geographical regions to general interference graphs.

## 2 Models and estimators

We now formalize the interference model using the potential outcomes framework [30]. Let $\mathbf{Z} \in \{-1, 1\}^N$ be an assignment of each of the $N$ experimental units to treatment ($Z = 1$) or control ($Z = -1$). The *potential outcome* of the $i^{th}$ experimental unit is denoted $Y_i$, and in the most general setting could be a function $Y_i(\mathbf{Z})$ of the entire treatment assignment vector. We further assume the existence of a (known) bipartite graph between experimental units and interference units, with nonnegative weights $w_{is} \geq 0$ encoding the relationship between experimental unit $i$ and interference unit $s$. Such a graph may be obtained from historical data on interactions between units $i$ and $s$, or on similarity between $i$ and $s$ measured by geography or other features [9, 31].

While there are many possible estimands of interest, our primary goal is to estimate the *average total treatment effect* $\tau = \frac{1}{N} \sum_{i \in [N]} Y_i(\mathbf{Z} = \mathbf{1}) - Y_i(\mathbf{Z} = -\mathbf{1})$, sometimes referred to as the average treatment effect or total treatment effect. Since we assign some units to treatment and others to control, it is impossible to observe any potential outcome under a fully treated ($\mathbf{Z} = \mathbf{1}$) or fully controlled ($\mathbf{Z} = -\mathbf{1}$) condition. If the underlying interference graph among units were composed of multiple connected components, so that only the assignments $Z$ in a unit's connected component affected its potential outcome, then it would be possible to assign treatment at the level of the connected component and observe fully treated or control outcomes. However, in realistic marketplaces, such a perfect separation almost never occurs. As a result, we require some modeling assumptions on the potential outcomes to infer the behavior of unit $i$ under the fully treated or controlled condition.

### 2.1 The potential outcome model

A popular approach to modeling potential outcomes with network interference is the *exposure mapping* paradigm [3, 15, 16]. An exposure mapping is a function $e_i : \{-1, 1\}^N \to \mathbb{R}$ such that $Y_i(Z_i, e_i(\mathbf{Z})) = Y_i(\mathbf{Z})$. In other words, the indirect effect of $Z_{j \neq i}$ on unit $i$ is captured completely by the exposure $e_i(\mathbf{Z})$. When it is clear from context, we will write $e_i$ instead of $e_i(\mathbf{Z})$ to denote the exposure of unit $i$. Given an exposure mapping, we can further posit a model for the effect of assignment $Z_i$ and exposure $e_i$ on the outcome $Y_i$. We principally consider the linear model

$$Y_i(\mathbf{Z}) = \alpha_i + \beta_i Z_i + \gamma_i e_i, \tag{1}$$

which is commonly used in the interference literature [6, 19], although other models are discussed in Section 4. Following the tradition of the finite population model, first defined by Neyman [2], we treat the coefficients $\alpha_i$, $\beta_i$ and $\gamma_i$ as fixed but unknown so that the only randomness in the observation model is due to the choice of treatment assignment $\mathbf{Z}$. This contrasts with alternatives such as a model in which coefficients are drawn from some super-population or a model in which the coefficients are common across $i$ with some additive error term $\varepsilon$ in the linear model [25]. The finite population model avoids making assumptions about a population from which experimental units are drawn, and for this reason is often preferred in the causal inference literature [32, 33].

### 2.2 The exposure model

The literature on network interference typically defines the exposure $e_i$ as a function of the treatments of the neighbors of $i$ [3, 15, 16]. For example, the exposure mapping might be the (weighted) fraction of neighbors that are treated [17], or the count of neighbors that are treated [26]. In the bipartite setting, experimental units are never immediate neighbors. Instead, experimental units interact through their relationships with interference units. Defining an exposure mapping that is analogous to those used in the network setting requires an analogous definition of the "neighborhood" of an experimental unit, as well as the weight of the connections from a unit to each of its neighbors.

We propose a bipartite analogue to the neighborhood-based exposure mapping, composed of two parts: the *dose $d_s \in [-1, 1]$* of each interference unit $s \in [M]$ represents the weighted average of the treatment assignment $Z_i$ of each experimental unit in the neighborhood of $s$, while the *exposure*

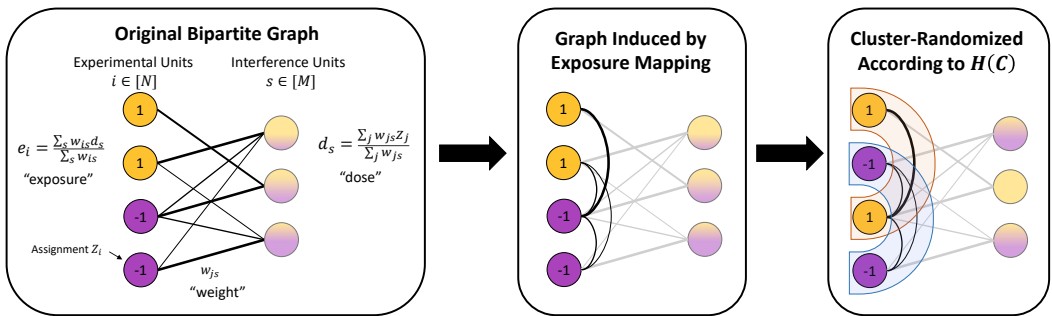

Figure 1: Overview of our cluster-randomized experimental design. Left Panel: We are given a bipartite graph connecting experimental units (left) with interference units (right) with edges of known weight $w$. Assigning treatments $Z_i \in \{-1, 1\}$ to each experimental unit induces a dose $d_s$ on each interference unit, and an exposure $e_i$ on each experimental unit. Middle Panel: The exposure of experimental unit $i$ depends on the assignment of other experimental units $j$, which induces a graph on the experimental units. Right Panel: Clustering the experimental units based on the induced graph creates clusters of units which heavily influence each others' outcomes. Randomizing the treatment assignment at the level of the cluster provides exposures $e_i$ that are much closer to the treatment $Z_i$ than would be achieved with a unit-randomized design.

$e_i \in [-1, 1]$ of each experimental unit $i \in [N]$ represents the weighted average of doses among the interference units in the neighborhood of $i$:

$$d_s = \frac{1}{\sum_{i \in [N]} w_{is}} \sum_{i \in [N]} w_{is} Z_i, \qquad e_i = \frac{1}{\sum_{s \in [M]} w_{is}} \sum_{s \in [M]} w_{is} d_s. \qquad (2)$$

See Figure 1 for an illustration of the exposure mapping. Because the exposure thus defined can be written as a linear combination of the treatment assignments of the two-hop neighbors of unit $i$, our definition of exposure can be viewed as an exposure mapping where $e_i$ depends on two-hop neighbors, in which we must impute the effective "assignment" of interference units $s$ from the assignments of their neighbors. Depending on the problem instance, it may be more appropriate to write one or both of these terms as an unnormalized linear combination instead of a convex combination; we show in Appendix A that all of our results apply to the unnormalized setting as well.

## 2.3 Estimators

Estimators for the average total treatment effect $\tau$ under network interference typically fall into one of two categories: difference-in-means (DIM) estimators and inverse propensity score (IPS) estimators. The former reports the difference between the mean of $N_T$ treated units and the mean of $N_C$ control units, where we treat $N_T$ and $N_C$ as fixed quantities chosen before treatment randomization occurs. The latter requires a notion of which units are "fully exposed" to treatment or to control, perhaps defined by $e_i$, and reweighs fully exposed observations by the inverse probability of achieving that state.

$$\widehat{\tau}_{DIM} = \sum_{Z_i = 1} \frac{Y_i}{N_T} - \sum_{Z_i = -1} \frac{Y_i}{N_C}, \quad \widehat{\tau}_{IPS} = \frac{1}{N} \sum_{i \in [N]} \frac{Y_i \mathbf{1}\{i \text{ fully treated}\}}{\mathbb{P}(i \text{ fully treated})} - \frac{Y_i \mathbf{1}\{i \text{ fully controlled}\}}{\mathbb{P}(i \text{ fully controlled})}$$

The difference in means estimator can suffer from bias in the presence of interference, because both the treated and control means may be biased estimates for their population quantities $n^{-1} \sum_i Y_i(\mathbf{1})$ and $n^{-1} \sum_i Y_i(-\mathbf{1})$, respectively. The IPS estimator is unbiased as long as a "fully treated" unit does in fact behave like a unit in which $\mathbf{Z} = \mathbf{1}$ (and likewise for control), however, this comes at a cost of high variance if the propensity scores are small. In our linear model, full exposure only occurs when $e_i = Z_i$, which requires all experimental units in the two-hop neighborhood of $i$ to have the same assignment as $i$. In many realistic bipartite graphs the chance of full exposure for a given unit is very low, increasing the bias in the IPS estimator to unacceptable levels. Appendix B illustrates the high variance of the IPS estimator for a selection of simulated bipartite graphs and a variety of definitions of "full treatment." The difference-in-means estimator also has the advantage

that it outperforms the IPS estimator in the setting of no interference, $Y_i(\mathbf{Z}) = Y_i(Z_i)$, since in this setting $\widehat{\tau}_{DIM}$ is unbiased and has the same or lower variance than $\widehat{\tau}_{IPS}$ for any definition of full exposure. As a result, the difference in means estimator can be seen as an "optimistic" choice of estimator, which makes better use of the data in the case of no interference while incurring bias in the presence of interference. For the reasons described above, we focus on the difference in means estimator in this work.

# 3 Experimental design

Having chosen the linear potential outcomes model of Equation (1) with the exposure mapping of Equation (2) and the difference-in-means estimator $\widehat{\tau}_{DIM}$, we seek a mechanism for randomly assigning experimental units to treatment or control that achieves low mean squared error (MSE) in recovering the average total treatment effect $\tau$. The MSE can be decomposed into the bias and the variance, where the former is caused by interference and the latter is primarily a function of the number of units randomized to treatment and control. Following previous works [26, 17, 27], we consider the class of *balanced cluster-randomized designs*, formalized in Definition 3.1.

**Definition 3.1** (Balanced $K$-cluster randomized design). *Let $\mathcal{C} = \{C_\ell\}_{\ell=1}^K$ be a partition of the $N$ experimental units into $K$ equally sized clusters. A balanced $K$-cluster randomized design $\mathcal{D}(\mathcal{C})$ is a distribution over vectors $\mathbf{Z} \in \{-1, 1\}^N$ generated by assigning $Z_i = 1$ for all experimental units $i$ belonging to a set of $K_T \in (0, K)$ clusters chosen uniformly at random among all $K$ clusters.*

If the clustering faithfully captures patterns of interference, cluster randomization can increase the chance that a treated (resp. control) unit is highly exposed to treatment (resp. control), thereby reducing the bias in $\widehat{\tau}_{DIM}$.

We control the variance of the estimator by enforcing a balanced clustering, so that assigning a fixed fraction of clusters to treatment results in a consistent fraction of units assigned to treatment. Since cluster randomization can change the effective number of experimental units [9, Section 4.2], the variance of $\widehat{\tau}_{DIM}$ will depend on the number of clusters chosen. In practice, the variance under a given clustering can typically be estimated using historical data, often called an A/A test. Practitioners can use these estimates to control the variance in light of the anticipated effect sizes. The bias due to interference, however, is not estimable with historical data when all such data is observed under the control condition. Balanced clustering also has practical implementation benefits, which we discuss further in Appendix E.

Since the variance can be estimated from historical data and controlled by the number of balanced clusters, we choose to focus on identifying a clustering that minimizes the bias of $\widehat{\tau}_{DIM}$ given a fixed number of balanced clusters. Choosing the correct number of clusters to trade off bias and variance remains an important direction for future work.

## 3.1 Existing designs do not work in the one-sided bipartite setting

A natural question is whether existing approaches to cluster-randomized designs are adequate for our setting of one-sided experiments on bipartite graphs. In this section, we consider the natural extensions of two clustering algorithms from the literature to our setting and describe failure modes for each of them.

**Direct clustering of the bipartite graph.** Cluster-randomized designs have primarily been studied in the context of graphs in which all units are experimental units, such as social networks [26, 17]. Within the class of cluster-randomized designs, balanced partitioning has been suggested as a means to minimize bias [17]. The natural analog in the bipartite setting is to create a balanced partitioning of the bipartite graph according to the provided edge weights $w_{is}$; this extension was considered as a baseline in a different bipartite setting by Pouget-Abadie et al. [18]. Clusters may contain both experimental units and interference units, but only experimental units count toward the balancedness constraint.

Unfortunately, this naive approach fails to take into account the two-hop structure of interference in a bipartite graph. Consider for example two experimental units $i$ and $j$ that must be assigned to a different cluster than that of their common neighbor, interference unit $s$, perhaps to satisfy a balancedness or cluster cardinality constraint. There is no benefit for the balanced partitioning

algorithm on the bipartite graph to assign these two experimental units to the same cluster as each other, despite them sharing a common neighbor. This mode of failure is illustrated with an example in Appendix C.1.

**Maximizing the variance of the doses.** Cluster-randomized designs have also been considered in the setting of two-sided bipartite experiments, in which treatment is assigned to the experimental units but outcomes are measured on the interference units. When the potential outcome of an interference unit is a linear function of the dose $d_s$, Pouget-Abadie et al. [18] and Harshaw et al. [19] recommend assigning treatments $\mathbf{Z}$ in a way that maximizes the empirical variance of the realized doses. In the one-sided bipartite setting it is important to enforce balancedness of the clusters to control the variance of $\widehat{\tau}_{DIM}$, so an extension of their objective to this setting would be to maximize $\mathrm{Tr}(\mathrm{Var}(\mathbf{d}))$ over a balanced cluster-randomized assignment.

Unfortunately, this clustering objective fails in the one-sided bipartite setting. It is possible that the doses $d_s$ of the interference units are primarily controlled by a small number of experimental units with especially large weights. In this case, ensuring that the doses are close to $-1$ or $1$ only translates into guarantees about those highly-weighted experimental units, and not necessarily about the typical experimental unit. This idea is illustrated with an example in Appendix C.2.

## 3.2 The bias-minimizing clustering objective $\mathcal{H}(\mathcal{C})$

Having shown that these two natural extensions fail to identify the bias-minimizing cluster-randomized designs, we turn our attention to a clustering objective $\mathcal{H}(\mathcal{C})$ that directly minimizes the bias in $\widehat{\tau}_{DIM}$.

**Lemma 3.2.** *Suppose the difference in means estimator $\widehat{\tau}_{DIM}$ is computed on a balanced $K$-cluster design $\mathcal{D}(\mathcal{C})$. Let the potential outcomes follow the linear model* (1). *Then the bias is given by:*

$$\tau - \mathbb{E}_{\mathbf{Z}\sim\mathcal{D}(\mathcal{C})}[\widehat{\tau}_{DIM}] = \frac{2}{N} \cdot \frac{K}{K-1} \sum_{i\in[N]} \sum_{j\notin\mathcal{C}(i)} \gamma_i \sum_{s\in[M]} \frac{w_{is}}{\sum_s w_{is}} \frac{w_{js}}{\sum_k w_{ks}}.$$

*Suppose further that all interference terms are bounded in magnitude by a constant, so that $\gamma_i = O(1)$. Then the minimax bias is bounded:*

$$\arg\min_{\mathcal{C}} \max_{\gamma:\ \gamma_i=O(1)} \left|\tau - \mathbb{E}_{\mathbf{Z}\sim\mathcal{D}(\mathcal{C})}[\widehat{\tau}_{DIM}]\right| = \arg\min_{\mathcal{C}} \sum_{i\in[N]} \sum_{j\notin\mathcal{C}(i)} \sum_{s\in[M]} \frac{w_{is}}{\sum_s w_{is}} \frac{w_{js}}{\sum_k w_{ks}}$$

$$=: \arg\min_{\mathcal{C}} \mathcal{H}(\mathcal{C}).$$

To interpret Lemma 3.2, we can consider the cases in which the clustering $\mathcal{C}$ can achieve zero bias for $\widehat{\tau}_{DIM}$. One way this can occur is when there is no interference, so that $\gamma_i = 0$ for all $i$. Even in the presence of interference, the bias of $\widehat{\tau}_{DIM}$ can be zero if the cut edges $w_{is}\ :\ j\notin\mathcal{C}(i)$ are all of zero weight. In such a well-clustered graph, the outcome of unit $i$ depends only on the treatment assignments of other units within its own cluster. Since the same treatment is applied to each element of a given cluster, a perfectly clustered graph has $e_i = Z_i$ and experiences no bias due to interference.

Our objective $\mathcal{H}(\mathcal{C})$ has two interpretations: as a clustering objective on an induced graph on experimental units, and as a statistical objective on the covariance between assignments and exposures.

**A graphical interpretation of $\mathcal{H}(\mathcal{C})$.** Interference occurs when the treatment assignment of unit $j$ affects the exposure of unit $i$. The influence of $Z_j$ on $e_i$ under the linear interference model (1) is

$$e_i(\mathbf{Z}_{j+}) - e_i(\mathbf{Z}_{j-}) = 2 \sum_{s\in[M]} \frac{w_{is}}{\sum_s w_{is}} \frac{w_{js}}{\sum_k w_{ks}}. \tag{3}$$

where $\mathbf{Z}_{j+}$ (resp. $\mathbf{Z}_{j-}$) is the vector $\mathbf{Z}$ with entry $Z_j$ set to $1$ (resp. $-1$). Now consider a directed graph on experimental units where the edge from $i$ to $j$ is weighted according to $e_i(\mathbf{Z}_{j+}) - e_i(\mathbf{Z}_{j-})$; clustering according to this graph minimizes the objective $\mathcal{H}(\mathcal{C})$. This is in fact a similar approach that was adopted in [9], which also looked at a similar folding of the graph and showed it to minimize the bias under a linear potential outcome model motivated by geographical migrations.

**A statistical interpretation of $\mathcal{H}(\mathcal{C})$.** A natural goal of a cluster-randomized design is to ensure that the exposure $e_i$ is close to the treatment $Z_i$. Lemma 3.3 shows that our clustering objective $\mathcal{H}(\mathcal{C})$ maximizes the covariance between the exposures and assignments.

**Lemma 3.3.** *Let $\mathcal{D}(\mathcal{C})$ be a balanced $K$-cluster randomized design. Then we have:* $\arg\max_{\mathcal{C}} Tr\left(Cov_{\mathbf{Z}\sim\mathcal{D}(\mathcal{C})}(\mathbf{Z}, \mathbf{e})\right) = \arg\min_{\mathcal{C}} \mathcal{H}(\mathcal{C})$.

Finally, we observe that our cluster-randomized design is an instance of the well-known *balanced partitioning* problem [34] on the graph over experimental units, with edges given by the explicit formula (3). Balanced partitioning is NP-hard, even in its relaxed form which enforces only partial balancedness, but many tools exist to compute it approximately [34, 35, 36].

## 4    Robustness

So far we have shown that a cluster-randomized design with objective $\mathcal{H}(\mathcal{C})$ minimizes the bias of $\widehat{\tau}_{DIM}$ under the normalized exposure mapping in by Equation (2) and the linear potential outcomes model in Equation (1). A natural question is: how robust is this design to deviations from this model? In this section we analyze the robustness of the design under alternative potential outcomes models, showing that the design remains minimax optimal for Lipschitz potential outcomes and minimizes an upper bound on the bias for a class of potential outcomes models motivated by the exposure mapping literature. Additionally, there may exist experimental settings where the exposure $e_i$ is better modeled without the normalization constant $\sum_s w_{is}$, such as when the outcome $Y_i$ is proportional to the edge weight incident to unit $i$. In this case, or in the case that the dose $d_s$ is unnormalized, all of our results remain valid for a generalization of the clustering objective which removes the corresponding normalization constant(s) from $\mathcal{H}(\mathcal{C})$. This idea is formalized in Appendix A.

### 4.1    Lipschitz potential outcomes

Let $Y_i(Z, e) \in \mathrm{Lip}_L(e)$ be a Lipschitz function in the exposure $e$, with Lipschitz constant $L$. Then we can bound the bias in $\widehat{\tau}_{DIM}$ by a multiple of $\mathcal{H}(\mathcal{C})$, and this bound is tight in a minimax sense.

**Lemma 4.1.** *Let observations $Y_i$ be observed from a $K$-cluster design, with potential outcomes given by $Y_i(Z, e) \in Lip_L(e)$. Then the bias of the difference-in-means estimator $\widehat{\tau}_{DIM}$ is bounded above by*

$$|\mathbb{E}[\widehat{\tau}_{DIM}] - \tau^*| \leq \frac{2}{N}\frac{K}{K-1}L \cdot \mathcal{H}(\mathcal{C}).$$

*Furthermore, this bound is tight over the class of Lipshitz functions, so that balanced clustering according to the objective $\mathcal{H}(\mathcal{C})$ is tight in a minimax sense among all balanced clusterings:*

$$\arg\min_{\mathcal{C}} \max_{f \in Lip_L(e)} |\mathbb{E}[\widehat{\tau}_{DIM}] - \tau^*| = \arg\min_{\mathcal{C}} \mathcal{H}(\mathcal{C}).$$

We note that the objective $\mathcal{H}(\mathcal{C})$ is minimax optimal over the class of $L$-Lipschitz functions regardless of the Lipschitz constant $L$. As a result, the practitioner need not know this constant in order to find the minimax optimal clustering.

### 4.2    Functions constant in neighborhoods of $\{-1, 1\}$

A common assumption in the exposure mapping literature is that units with exposure $e_i$ close enough to their treatment assignment $Z_i$ behave as if their entire neighborhood were assigned to $Z_i$. This assumption motivates the use of IPS estimators, which are unbiased in that setting. If we let $\Delta$ denote the neighborhood of $Z_i$ in which an exposure is considered fully treated or controlled, then we have the following constraint on the potential outcome function.

$$|Y_i(Z, e) - Y_i(Z, Z)| \begin{cases} = 0 & \text{if } |Z - e| < \Delta \\ \leq B & \text{otherwise} \end{cases} \qquad \forall Z \in \{-1, 1\}, \ \forall e \in [-1, 1] \qquad (4)$$

Under this assumption on the behavior of $Y_i$, we can upper bound the bias of the difference-in-means estimator by a quantity that will turn out to be minimized by minimizing $\mathcal{H}(\mathcal{C})$.

**Lemma 4.2.** *Let observations $Y_i$ be observed from a $K$-cluster design, with potential outcomes satisfying Equation (4). Then the bias of the difference-in-means estimate $\widehat{\tau}_{DIM}$ is bounded above by*

$$|\mathbb{E}[\widehat{\tau}_{DIM}] - \tau^*| \leq \frac{2B}{N\Delta}\frac{K}{K-1}\mathcal{H}(\mathcal{C}).$$

Table 1: Relative bias of $\hat{\tau}_{DIM}$ as the bipartite stochastic block model changes (see 5.1)

|  | $p = 0.0$ | $p = 0.005$ | $p = 0.05$ | $p = 0.5$ |
|---|---|---|---|---|
| $\mathcal{H}(\mathcal{C})$ | **0.02($\pm$0.06)** | **3.90($\pm$0.06)** | **11.54($\pm$0.05)** | 12.98($\pm$0.08) |
| Tr(Var(d)) objective | **0.01($\pm$0.06)** | **3.84($\pm$0.05)** | **11.49($\pm$0.06)** | 12.91($\pm$0.07) |
| Direct clustering | **0.01($\pm$0.05)** | 9.10($\pm$0.13) | 12.68($\pm$0.07) | 12.95($\pm$0.07) |
| EXPOSURE-DESIGN | 0.33($\pm$0.06) | 4.06($\pm$0.06) | 11.90($\pm$0.07) | 13.00($\pm$0.08) |
| Unit-level randomization | 12.44($\pm$0.08) | 12.55($\pm$0.08) | 12.76($\pm$0.08) | 12.95($\pm$0.08) |
| True clusters | 0.01($\pm$0.06) | 3.88($\pm$0.06) | 11.58($\pm$0.06) | 12.96($\pm$0.06) |

Lemma 4.2 presents only an upper bound on the bias, and in general this bound is not tight. However, we provide simulations in Section 5.2 showing that the objective $\mathcal{H}(\mathcal{C})$ is a reasonable heuristic for functions that satisfy Equation (4).

# 5 Experiments

We explore the performance of our cluster-based randomized design in several settings using simulated graphs. We compare to the baseline of unit-level randomization as well as cluster-level randomization according to several clustering schemes: the true clustering, direct clustering on the original bipartite graph (Section 3.1), maximizing the Tr(Var(**d**)) objective (Section 3.1), and maximizing the expected empirical variance of the doses as motivated by [18, 19]. We use code provided by the authors of [35] to identify a minimum-cost balanced partitioning of the graphs induced by the objectives $\mathcal{H}(\mathcal{C})$, Tr(Var(**d**)), and the direct clustering. For the last objective, which is not a balanced partitioning, we use code provided by the authors of [19] to minimize the EXPOSURE-DESIGN objective, reporting results with the hyperparameter $\phi$ tuned to minimize mean squared error. See Appendix H for an overview of the algorithms.

## 5.1 Robustness to clusterability in the stochastic block model

We begin by studying the performance of our design as the amount of interference varies. We construct a synthetic graph according to the bipartite stochastic block model with $N = 1,000$ experimental units and $M = 2,000$ interference units. Both sides of the graph are partitioned into 20 equally sized groups with label $i \in [20]$. Experimental and interference units with the same label have an edge of weight 1 with probability 0.5, while units with different labels have an edge of weight 1 with probability $p$. We experiment with values from $p = 0$ (no interference between clusters) to $p = 0.5$ (the absence of an underlying clustering structure). Potential outcomes were drawn according to the linear model (1), with coefficients drawn $\alpha_i \sim \mathcal{N}(0, 1)$, $\beta_i \sim \mathcal{N}(1, 1)$, and $\gamma_i \sim \mathcal{N}(-1, 1)$. All clustering designs used $K = 20$ clusters, with $K_T = 10$ clusters assigned to treatment. Table 1 shows the relative bias (defined as $|\mathbb{E}[\hat{\tau}] - \tau|/\tau$) of $\hat{\tau}_{DIM}$ under each cluster-randomized design; variance is inconsequential in this setting, so the variance and MSE are reported in the supplementary materials. Uncertainty represents the 95% confidence interval as determined by bootstrapping over 100 random draws of treatment assignment **Z**, over a single draw of the graph and potential outcomes.

As anticipated, all designs incur lower bias when there is less interference in the graph (i.e., when $p$ is smaller). Our clustering objective $\mathcal{H}(\mathcal{C})$ and the Tr(Var(**d**)) objective perform on par with the true clustering for all values of $p$. The low bias of the $\mathcal{H}(\mathcal{C})$ objective is unsurprising given the minimax optimality result in Lemma 3.2. The equivalent performance of the Tr(Var(**d**)) objective is due to the symmetry inherent in the bipartite stochastic block model, in which each interference unit has the same incoming edge weight $\sum_i w_{is}$ in expectation, and the same is true of the experimental units' edge weights $\sum_s w_{is}$. In such a symmetric setting the objectives $\mathcal{H}(\mathcal{C})$ and Tr(Var(**d**)) are equivalent, as can be seen by comparing their representations in Lemma 3.2 and Equation (5).

## 5.2 Robustness to nonlinearity

Next we explore the robustness of our design to nonlinearity in the potential outcomes model $Y_i$ by simulating outcomes according to Equation (4), in which the potential outcome is constant for $e_i$ in

Table 2: Relative bias of $\widehat{\tau}_{DIM}$ as the neighborhood of pure exposure, $\Delta$, widens (see 5.2)

|  | $\Delta = 0.1$ | $\Delta = 0.3$ | $\Delta = 0.5$ |
|---|---|---|---|
| $\mathcal{H}(\mathcal{C})$ | 1.000($\pm$0.004) | **0.457($\pm$0.005)** | **0.001($\pm$0.000)** |
| Tr(Var(d)) objective | 1.002($\pm$0.004) | **0.460($\pm$0.004)** | **0.000($\pm$0.000)** |
| Direct clustering | 0.997($\pm$0.005) | 0.950($\pm$0.008) | 0.600($\pm$0.020) |
| EXPOSURE-DESIGN | 1.001($\pm$0.004) | 0.509($\pm$0.005) | 0.009($\pm$0.001) |
| Unit-level randomization | 0.998($\pm$0.004) | 1.000($\pm$0.004) | 0.998($\pm$0.003) |
| True clusters | 1.001($\pm$0.004) | 0.458($\pm$0.004) | 0.001($\pm$0.000) |

a $\Delta$ neighborhood of $Z_i$. We simulate $Y_i = -Z_i$ if $|e_i - Z_i| < \Delta$ and $Y_i \sim \mathcal{U}(-1, 1)$ otherwise, encoding a setting in which no knowledge can be gained about $\tau$ when $e$ is $\Delta$-far from $Z$. The graph is given by the bipartite stochastic block model described in Section 5.1 with 20 groups, with $p = 0.5$ chance of an edge between units belonging to the same group and $p = 0.005$ for units belonging to different groups. All clustering designs used $K = 20$ clusters, with $K_T = 10$ clusters assigned to treatment. Table 2 shows the results of these experiments. Uncertainty represents the 95% confidence interval as determined by bootstrapping over 100 random draws of treatment assignment $\mathbf{Z}$, over a single draw of the graph and potential outcomes.

We see that all designs have lower bias when $\Delta$ is large, reflecting the fact that observations with exposure $e_i$ $\Delta$-far from $Z_i$ are useless in determining $\tau$, and it is easier to get $\Delta$-pure observations for large $\Delta$. The results in Table 2 support our claim that $\mathcal{H}(\mathcal{C})$ is a reasonable design heuristic for this setting (motivated by the bias upper bound in Lemma 4.2). We observe again that the $\text{Tr}(\text{Var}(\mathbf{d}))$ objective does as well as the $\mathcal{H}(\mathcal{C})$ objective; as described in Section 5.1, this is due to the fact that these objectives are nearly equivalent under the symmetry of our simulated graphs.

### 5.3 Performance on power-law graphs

To contrast with the bipartite stochastic block models studied above, we experiment on a bipartite graph model with a power-law distribution of the vertices. Our graph model combines the bipartite preferential attachment model of [37] with the affinity model of [38]. In this model, each experimental unit is assigned to one of $K$ latent classes. For each experimental unit $i$, a degree $d_i$ is drawn from the power-law distribution $d_i = 2X \; : \; X \sim \text{Zipf}(3)$. For each edge $j \in [d_i]$, it is either attached to an existing interference unit with probability $1 - \lambda$, or to a newly drawn interference unit with probability $\lambda$. If attached an existing unit, unit $s$ is chosen with probability proportional to $(d_s + p)$ if $s$ is of the same latent class as $i$, or $d_s + q$ if $s$ is of a different class. If a new unit is drawn, then it is of the same class as unit $i$ with probability $p(p + (K-1)q)^{-1}$, and of each other class with probability $q(p + (K-1)q)^{-1}$. We chose $N = 100$, $K = 10$, $\lambda = 0.5$, $q = 0.02p$. We experimented with both $p = 0.1$ (weak latent structure) and $p = 100$ (strong latent structure).

Table 3 shows the relative bias of four estimators when the potential outcomes are drawn from the linear model given in Section 1. Interestingly, in this setting we see that the cluster-randomized designs outperform the true latent clusters; this is possible because the interference occurs with respect to a single draw of the random graph, which the clustering algorithm gets to see. We speculate that factors specific to the power-law graphs, notably the existence of vertices with very high degree, might cause the optimal clustering for a given draw of the graph to be very different from the optimal clustering for the graph on average. This could have negative consequences for experimental design when only a random draw of the edges is observed, but the interference occurs according to the underlying latent structure.

### 5.4 Robustness to exposure mapping: an Airbnb case study

So far we have only considered potential outcomes models where the outcome $Y_i$ is a function of the assignment $Z_i$ and the exposure $e_i$ as defined in Equation (2). However, in many real-life settings, an exposure model is only an approximation to the true mechanism of interference.

We test the robustness of our experimental design to misspecification of the exposure mapping by simulating outcomes according to a model developed for vacation rentals by Li et al. [10], in which

Table 3: Relative bias of $\widehat{\tau}_{DIM}$ under an affinity model with a power-law distribution (see 5.3)

| | Strong latent structure | Weak latent structure |
|---|---|---|
| $\mathcal{H}(\mathcal{C})$ | **1.9($\pm$0.1)** | **3.5($\pm$0.1)** |
| Tr(Var(d)) objective | **2.0($\pm$0.1)** | 3.7($\pm$0.1) |
| Direct clustering | **1.8($\pm$0.2)** | **3.3($\pm$0.2)** |
| Unit-level randomization | 4.9($\pm$0.1) | 5.4($\pm$0.1) |
| True clusters | 2.9($\pm$0.1) | 5.2($\pm$0.1) |

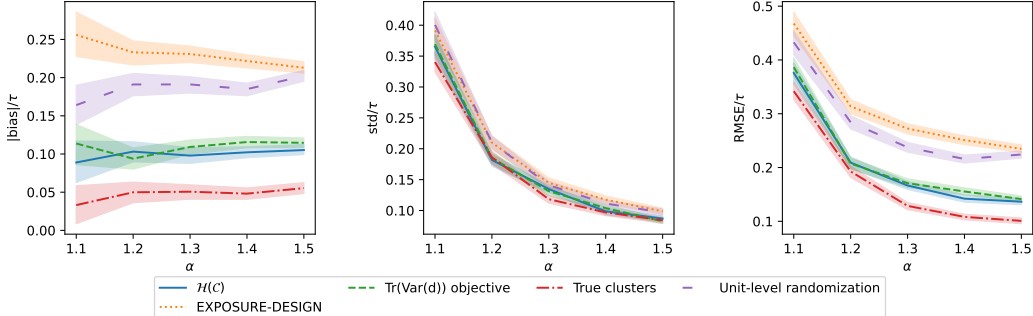

Figure 2: Performance of various clustering algorithms as the treatment effect $\alpha$ increases. Uncertainty represents 95% empirical confidence intervals over 500 draws of the treatment assignment **Z**. Direct balanced partitioning had error 4-8 times the second-worst algorithm, and is not shown.

there is no explicit exposure mapping defined. We call the experimental units *customers* and the interference units *listings*. In the first phase of the model, each customer $i$ applies to each listing $s$ with probability $\phi_{is}$. In the second phase, listings with applications randomly select an application to accept. The measured outcome $Y_i$ is 1 if customer $i$ successfully booked a listing, and 0 otherwise. In alignment with previous work in this literature [10, 28] we create a natural clustering structure on the network by assigning each customer and each listing to one of 20 types. Our simulation takes $N = 500$ customers and $M = 1000$ listings, with application probability under the control assignment of $\phi_{is} = 0.016$ if $i$ and $s$ are of the same type, and $\phi_{is} = 0.0001$ otherwise. Treating customer $i$ increases these application probabilities by a factor of $\alpha$.

When running this marketplace experiment, an experimenter would typically have access to historical data about the rate of successful applications of customer $i$ to listing $s$, but would not know the consideration probabilities $\phi_{is}$. To be faithful to this observation model, we constructed the bipartite graph using twelve rounds of interaction in this marketplace under the control condition. This graph was sampled once and fixed for all experiments. All clustering designs used $K = 20$ clusters, with $K_T = 10$ assigned to treatment. Simulations were performed by extending code provided by the authors of Li et al. [10] to the cluster-randomized setting. The relative bias, standard deviation, and root mean squared error (RMSE) of $\widehat{\tau}_{DIM}$ of various clustering algorithms are shown in Figure 2.

We observe that all clustering algorithms had similar variance but that the bias of $\mathcal{H}(\mathcal{C})$ and Tr(Var(**d**)) were closest to that of the true clustering, giving those two methods the lowest RMSE. Notably, the unit-randomized design is suggested by [10] in this setting ($N < M$), but our cluster-randomized design outperforms it. We conclude that our clustering method still outperforms other baselines, even in a setting where the potential outcome does not obey the exposure mapping model we hypothesized.

## Acknowledgments

The authors would like to thank Christopher Harshaw, Khashayar Khosravi, Kay Brodersen, Vahan Nanumyan, and Kevin Jamieson for helpful discussions; Hannah Li and Geng Zhao for sharing their code for the Airbnb simulator; David Eisenstat for assistance with the clustering code; and Robbie Weber for feedback on the paper organization.

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
