# A  Discussion of alternative normalizations of the exposure and dose models

Recall that, under the definition of exposure given in Equation (2), both the dose $d_s$ and the exposure $e_i$ are normalized by the sum of adjacent edges so that they lie in the range $[-1, 1]$. Depending on the nature of the experiment, it may instead be appropriate to define an exposure mapping in which either or both of these quantities are unnormalized, so that $\tilde{d}_s = \sum_i w_{is} Z_i$ and/or $\tilde{e}_i = \sum_s w_{is} d_s$. For example, suppose that the edge weights $w_{is}$ measure the value of goods purchased by buyer $i$ from seller $s$ and that the treatment results in a multiplicative increase in the cost per unit good. In this setting, a more appropriate model might be one with unnormalized exposure, for example $Y_i = \gamma_i \sum_s w_{is} d_s$ where $\gamma_i$ represents the multiplicative increase in the cost per unit good for customer $i$ under treatment (the individual treatment effect). This is the approach to normalization taken by Rolnick et al. [9], who study the bipartite model in the context of search queries issued by users (experimental units) across various geographical regions (interference units). Their potential outcomes model assumes a normalized dose but unnormalized exposure.

We note that we could also write this unnormalized exposure model in terms of the original linear model (1) by absorbing the exposure normalization term $(\sum_s w_{is})^{-1}$ into $\gamma_i$. However, if the normalization terms varied significantly between buyers $i$ while the individual treatment effects were approximately equal, then the minimax guarantee in Lemma 3.2 would be less meaningful when applied to the normalized model than the unnormalized exposure model. In general, the normalization of the dose and exposure should be chosen so that the magnitude of $\gamma_i$ in the linear model (1) has the least variance between experimental units $i$ as possible, as this is the setting in which the minimax result of Lemma 3.2 is the tightest.

We can define the generalized objective parameterized by the definitions of dose and exposure (normalized or unnormalized) using the notation of Equation (3):

$$\mathcal{H}_{e,d}(\mathcal{C}) = \sum_{i \in [N]} \sum_{j \notin \mathcal{C}(i)} \gamma_i \sum_s \tfrac{1}{2} \left( e_i(\mathbf{d}_{s+}) - e_i(\mathbf{d}_{s-}) \right) \cdot \tfrac{1}{2} \left( d_s(\mathbf{Z}_{j+}) - d_s(\mathbf{Z}_{j-}) \right)$$

where $\mathbf{d}_{s+}$ (resp. $\mathbf{d}_{s-}$) is the vector $\mathbf{d}$ with entry $d_s$ set to 1 (resp. $-1$), and analogously for $\mathbf{Z}_{j+}$ and $\mathbf{Z}_{j-}$. In the setting of normalized dose and response, this simplifies to the original objective $\mathcal{H}(\mathcal{C})$. Choosing the unnormalized dose $\tilde{d}$ yields the difference $\tilde{d}_s(\mathbf{Z}_{j+}) - \tilde{d}_s(\mathbf{Z}_{j-}) = 2w_{is}$, and choosing the unnormalized exposure $\tilde{e}$ yields $\tilde{e}_i(\mathbf{d}_{s+}) - \tilde{e}_i(\mathbf{d}_{s-}) = 2w_{js}$.

Under the linear potential outcomes model, the generalized objective satisfies the same properties as the original objective function: in particular, it is bias-minimizing in a minimax sense over all $\gamma_i \in [\Gamma_0, \Gamma_1]$, it minimizes $\text{Tr}(\text{Cov}(\mathbf{Z}, \mathbf{e}))$ for the given definition of $e$, and it has a graphical interpretation as minimizing the cut edges among an induced graph on experimental units. The first two properties can be verified via straightforward modifications of the proofs of Lemmas 3.2 and 3.3, respectively, while the last property can be seen from the analogue of Equation (3) in the unnormalized setting.

We conclude by discussing the relationship between the $\text{Tr}(\text{Cov}(\mathbf{Z}, \mathbf{e}))$ objective (ours) and the $\text{Tr}(\text{Var}(\mathbf{d}))$ objective (of [18, 19], discussed in Section 3.1), which turn out to differ only in the normalization terms. We first observe that the $\text{Tr}(\text{Var}(\mathbf{d}))$ objective can be written in a form similar to that of $\mathcal{H}(\mathcal{C})$, but with a different normalization on $w_{is}$:

$$\arg\max_{\mathcal{C}} \text{Tr}\left( \text{Var}_{\mathbf{Z} \sim \mathcal{D}(\mathcal{C})}(\mathbf{d}) \right) = \arg\min_{\mathcal{C}} \sum_{i \in [N]} \sum_{j \notin \mathcal{C}(i)} \sum_{s \in [M]} \frac{w_{is}}{\sum_k w_{ks}} \frac{w_{js}}{\sum_k w_{ks}}. \tag{5}$$

In the special case that the exposure normalization term $\sum_s w_{is}$ is equal to the dose normalization term $\sum_k w_{ks}$ for all $i$ and $s$, these constants drop out of the $\arg\min$ and the $\text{Tr}(\text{Var}(\mathbf{d}))$ objective becomes equivalent to the $\mathcal{H}(\mathcal{C})$ objective. This is approximately the setting of our synthetic experiments, in which all experimental units had the same expected number of connections (and the same for the interference units), which helps to explain why the $\text{Tr}(\text{Var}(\mathbf{d}))$ was so competitive in our experiments.

# B  High variance of the IPS estimator

In this section we substantiate the claim from Section 2.3 that the inverse propensity score (IPS) estimator is too high-variance to be practical in the settings we consider. The core idea is that, when

we run one-sided bipartite experiments, the definition of "complete exposure" to treatment or control is defined based on a unit's two-hop neighborhood. In a graph with even moderate connectivity, these neighborhoods are so large that units have a very low chance of being fully exposed to treatment or control. These small probabilities of full exposure in turn lead to high variance of the IPS estimator.

We begin by observing that, using cluster-randomized designs as we have defined them in 3.1, there is essentially zero chance of each unit being connected completely to treatment or control under even moderate connectivity. To see this, we will compute the chance of a given experimental unit $i$ being a two-hop neighbor of unit $j$ under the bipartite stochastic block model discussed in Section 5.1. Recall that in this model, $N$ experimental units and $M$ interference units are each partitioned into $K$ equal-sized clusters labeled $1, \ldots, K$. Units with the same label have an edge between them with probability $q = 0.5$, while units with different labels have an edge with probability $p$. We can compute the chance of experimental unit $i$ **not** having a two-hop neighbor in $\mathcal{C}(j)$ if $\mathcal{C}(i) \neq \mathcal{C}(j)$ is given by

$$\mathbb{P}(i \text{ has no 2-hop neighbor in } \mathcal{C}(j) | \mathcal{C}(i) \neq \mathcal{C}(j))$$

$$= \prod_{k \in \mathcal{C}(j)} \prod_{s \in \mathcal{C}(j)} \mathbb{P}(k \text{ not connected to } s \text{ OR } i \text{ not connected to } s) \cdot$$

$$\prod_{k \in \mathcal{C}(j)} \prod_{s \in \mathcal{C}(i)} \mathbb{P}(k \text{ not connected to } s \text{ OR } i \text{ not connected to } s)$$

$$= \prod_{k \in \mathcal{C}(j)} \prod_{s \in \mathcal{C}(j)} (1 - pq) \cdot \prod_{k \in \mathcal{C}(j)} \prod_{s \in \mathcal{C}(i)} (1 - pq)$$

$$= (1 - pq)^{2MN/K^2}$$

The number of clusters to which unit $i$ is connected, besides $\mathcal{C}(i)$, is the binomial random variable $\text{Binom}\left(K - 1, 1 - (1 - pq)^{2MN/K^2}\right)$. Observe that the probability of connection grows very quickly. In the setting of our experiments in Section 5.1, we have $M = 2000$, $N = 1000$, $q = 0.5$ and $K = 20$, so that when $p = 0.005$ we have $\mathbb{P}(i \text{ has a two-hop neighbor in } \mathcal{C}(j) | \mathcal{C}(i) \neq \mathcal{C}(j)) = 1 - 1 \cdot 10^{-11}$, i.e. each unit is almost certain to have a two-hop neighbor in every other cluster. We conclude that, in the graph settings we study, experimental units are likely to be connected to units of all clusters, making pure exposure to treatment or control an exceedingly rare occurrence.

To further illustrate this point, we compute the variance of the IPS estimator on our simulated graphs for various definitions of "pure exposure". Under Bernoulli cluster randomization (which is slightly different from the design studied in the rest of the paper, but which is standard for the IPS estimator) we have:

$$\text{Var}(\widehat{\tau}_{IPS}) = \frac{1}{N^2} \sum_{i \in [N]} \frac{1}{\mathbb{P}(\text{unit } i \text{ is treated})} Y_{i,T}^2 + \frac{1}{\mathbb{P}(\text{unit } i \text{ is controlled})} Y_{i,C}^2 \tag{6}$$

where $Y_{i,T}$ is the value of $Y_i$ under full treatment, and $Y_{i,C}$ is the value of $Y_i$ under full control. We compute the probabilities of full treatment and control via Monte Carlo simulation over draws of $\mathbf{Z}$ from a Bernoulli randomized design.

We compute the variance of the IPS estimator in two settings studied in Section 5.

Table 4 compares the bias of $\widehat{\tau}_{DIM}$ to the standard deviation of $\widehat{\tau}_{IPS}$ in the setting of Section 5.2, in which units behave as if they were fully exposed whenever $|Z_i - e_i| < \Delta$. This is precisely the setting in which the $\widehat{\tau}_{IPS}$ estimator works best, and it is unbiased in this case. We see from the table that the IPS estimator has less error than $\widehat{\tau}_{DIM}$ for small values of $\Delta$, but that when $\Delta = 0.5$, the bias of $\widehat{\tau}_{DIM}$ has decreased to be lower than the IPS variance. We note that these figures were generated for the graph with $p = 0.005$; increasing the connectivity (say, to $p = 0.05$) further increases the error of $\widehat{\tau}_{IPS}$ relative to $\widehat{\tau}_{DIM}$.

Table 5 compares $\widehat{\tau}_{IPS}$ and $\widehat{\tau}_{DIM}$ in the linear setting of Section 5.1. In this setting, the IPS estimator relies on the incorrect assumption that units in a $\Delta$ neighborhood of pure exposure act as if they were purely exposed to treatment or control; this is untrue in the linear model, where even slight exposure to the opposite treatment results in a change in $Y_i$. As a result, the standard deviation reported in Table 5 provides a lower bound on the RMSE of $\widehat{\tau}_{IPS}$, with the remainder of the error due to bias. We see that the IPS estimator with $\Delta = 0.1$ and $\Delta = 0.3$ has relative error that is much higher than

Table 4: RMSE (relative to $\tau$) of $\widehat{\tau}_{DIM}$ and $\widehat{\tau}_{IPS}$ as the neighborhood of pure exposure, $\Delta$, widens (see 5.2)

|  | $\Delta = 0.1$ | $\Delta = 0.3$ | $\Delta = 0.5$ |
|---|---|---|---|
| $\widehat{\tau}_{DIM}$ | 1.001 | 0.459 | 0.001 |
| $\widehat{\tau}_{IPS}$ | 0.429 | 0.043 | 0.032 |

Table 5: Bias (relative to $\tau$) of $\widehat{\tau}_{DIM}$ and standard deviation (relative to $\tau$) of $\widehat{\tau}_{IPS}$ as the bipartite stochastic block model changes (see 5.1)

|  | $p = 0.0$ | $p = 0.005$ | $p = 0.05$ | $p = 0.5$ |
|---|---|---|---|---|
| $\widehat{\tau}_{DIM}$ | 0.01 | 3.88 | 11.58 | 12.96 |
| $\widehat{\tau}_{IPS}(\Delta = 0.1)$ | 0.70 | 9.88 | 390.63 | 508.78 |
| $\widehat{\tau}_{IPS}(\Delta = 0.3)$ | 0.70 | 9.5 | 13.90 | 16.54 |
| $\widehat{\tau}_{IPS}(\Delta = 0.5)$ | 0.70 | 0.71 | 3.70 | 4.31 |

$\widehat{\tau}_{DIM}$, even before including the bias of $\widehat{\tau}_{IPS}$. When $\Delta = 0.5$, we expect the bias of $\widehat{\tau}_{IPS}$ to be substantial.

We conclude that, even though $\widehat{\tau}_{IPS}$ provides an unbiased estimate of $\tau$ when the exposure mapping is correctly specified, the variance in this estimator can be significant enough to justify using the biased estimator $\widehat{\tau}_{DIM}$.

## C   Counterexamples for alternative clustering methods

In this section, we further expand upon the claims of section 3.1 with explicit examples illustrating failures of direct balanced partitioning and of maximizing the variance among doses. Figure 3 illustrates the two counterexamples.

### C.1   Direct clustering of the bipartite graph

At a high level, the direct clustering approach fails to enforce clustering of the neighbors of interference unit $s$ that fall outside of cluster $\mathcal{C}(s)$. This mode of failure is illustrated in Figure 3a, where a direct clustering of the bipartite graph assigns the same objective function score to the two clusterings,

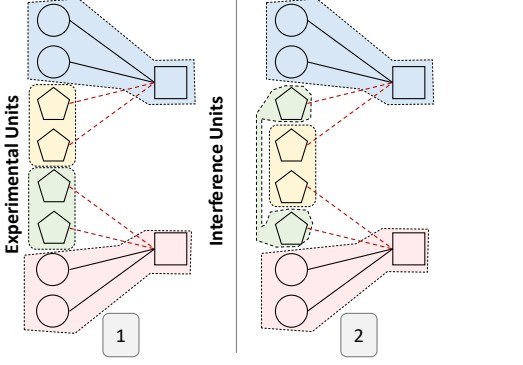

(a) **Failure of the direct balanced partitioning.**

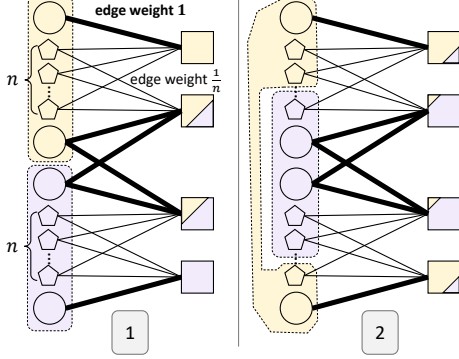

(b) **Failure of maximizing the dose variance.**

Figure 3: These counterexamples illustrate the inadequacy of existing cluster-randomized designs in the one-sided bipartite setting. In each case, Clustering 1 is the bias-minimizing clustering under our exposure model (2) and linear potential outcomes model (1).

even though the first is bias-optimal under our linear model while the second is not. Both clusterings have cost 4 according to direct partitioning (red dashed lines). We can use the linearity of $\widehat{\tau}_{DIM}$ to express the bias as a sum of bias contributions of each experimental unit (see Lemma F.1). The bias contributed by the circular units is the same in both clusterings, since the expected exposure impurity $|e_i - Z_i|$ is the same in both clusterings and our interference model is linear in $e_i$ and $Z_i$. However, the exposure of the pentagonal units in Clustering 1 stochastically dominates that of Clustering 2 in the sense of being closer to the true treatment assignment $Z_i$, which means that Clustering 1 incurs less bias. Therefore, Clustering 1 is superior under our potential outcomes model.

## C.2   Maximizing the variance of the doses

Figure 3b exhibits a bipartite graph in which the doses $d_s$ are primarily controlled by a small number of experimental units with highly weighted edges, so that maximizing $\mathrm{Tr}(\mathrm{Var}(\mathbf{d}))$ actually reduces the average covariance of $Z_i$ and $e_i$. Clustering 1 ensures that the majority of experimental units (the $2n$ pentagons) have $e_i = Z_i$ w.p. $\approx \frac{1}{2}$, and $e_i = \frac{3}{4}Z_i$ otherwise, while sacrificing the exposure of the two middle circular units, which have $e_i = 0$ w.p. $\approx \frac{1}{2}$. By contrast, Clustering 2 obtains a high dose variance by prioritizing the assignment of the middle circular units to the same cluster, at the expense of making $e_i = 0$ w.p. $\frac{1}{2}$ for all $2n$ pentagonal units. If we think of the interference among all units as being on the same order of magnitude then Clustering 1, which incurs less interference in the $2n$ pentagonal units, is the correct choice as $n \to \infty$. If, on the other hand, the potential outcomes scale in magnitude with the sum of an experimental unit's edge weights, then there are settings in which Clustering 2 is bias-optimal. This can be addressed by choosing the appropriate normalization of the dose and exposure for the problem at hand and modifying $\mathcal{H}(\mathcal{C})$ to reflect the choice of normalization; see Appendix A for further discussion.

# D   Comparison to two-sided designs

A recent development in the bipartite design literature is the two-sided randomization designs of Bajari et al. [29] and Johari et al. [28], in which experimental and interference units are each randomized to treatment and control, but treatment is only applied when a treated unit interacts with another treated unit. The major benefit of this design is that it allows for the quantification of spillover effects by comparing the fully controlled interactions (in which both sides of the interaction were controlled) to the interactions that experience interference (in which only one side of the interaction was assigned to control). In the absence of interference, these interaction types should have the same average response; any deviation from equality indicates interference that can be quantified.

The idea of measuring and correcting for interference is very appealing, especially when interference can be quantified using the same experiment that measures the global treatment effect. We think that this line of work is an important part of mitigating interference, and that the two-sided randomization design is a particularly clever way of approaching the problem. One challenge with TSR designs, as discussed by Bajari et al. [29, Section 6], is that without further assumptions on the interference model, TSR with treatment fraction $p$ is only able to estimate the global effect of treating $p$-fraction of the units as compared with treating no units, instead of the global effect of treating all units as compared with treating no units. Bajari et al. [29, Section 8.1] suggests randomizing at the interaction level to allow estimation of the spillover effect at multiple levels of interaction, which could then be extrapolated to estimate the global treatment effect - this of course relies on some model for the extrapolation. Ultimately, then, one difference between the TSR design and the cluster-randomized design is that the latter tries to get as many units in "pure treatment" and "pure control" as possible, to estimate the global treatment effect while limiting the need for such extrapolation. As Johari et al. [28, Section 8] suggests, cluster-randomization is likely to outperform TSR when the underlying graph is well-clusterable, but TSR is likely to do better when no cluster-based method can achieve near-complete treatment or control of units.

It would be interesting to think about combinations of TSR and cluster-based designs, particularly in the context of the extrapolation ideas proposed in Bajari et al. [29, Section 8.1]. One option would be to cluster-randomize the buyers and sellers individually, in a way that maximizes the variance of the unit-level exposures to treatment under a TSR design (echoing the objective of Pouget-Abadie et al. [18] and Harshaw et al. [19]). The TSR design would allow for the estimation of spillover

effects at various levels of interference, while maximizing the variance among the realized levels of interference would improve the accuracy of the extrapolation.

# E    Considerations for balanced clustering

In this appendix, we discuss potential benefits and detriments of cluster-randomized designs, expanding upon discussion in Section 3.

## E.1    Implementation benefits of balanced clusters

Practitioners may appreciate two benefits of balanced designs beyond the variance reduction mentioned in Section 3. First, when exactly $K_T$ of $K$ clusters are treated, balancing the clusters ensures control over the fraction of units that are treated. Controlling this fraction is important when we want to balance the scientific value of experimentation with potential negative effects on the treated units (i.e. staying within the experimental budget). Secondly, it has been our experience that many clustering algorithms that do not control for balancedness and cardinality sometimes produce many singletons clusters, when clustering incentives are not strong enough to group these units with other units. This is what we observed with the Exposure-Design objective of Harshaw et al. [19] when its hyperparameters are not tuned properly. When datasets are quite large, the large number of singleton clusters produced by these algorithms can slow down certain data analysis pipelines that work better in low cardinality settings. If these singletons clusters are to be clustered to reduce cardinality without improving any "cut"-like objective, it may make sense to do so in a balanced way for any of the reasons listed above. In other words, the occasional practical need to control for cluster cardinality is extra motivation to maintain balance instead of an arbitrary grouping of isolated nodes.

## E.2    Considerations in the presence of cluster-based heterogeneity

We studied a clustering objective, $\mathcal{H}(\mathcal{C})$, that minimizes the bias of the difference-in-means estimator for one-sided bipartite experiments. When the experimental units are well-clusterable, the cluster-randomized design significantly reduces the bias in $\widehat{\tau}_{DIM}$ when compared with unit-level randomization. However, cluster-based randomization may actually reduce the accuracy of the treatment effect estimate if the clustering correlates with individual treatment effects $\tau_i := Y_i(\mathbf{Z} = \mathbf{1}) - Y_i(\mathbf{Z} = -\mathbf{1})$. This may result in significant heterogeneity in the average treatment effect among clusters, which introduces additional variance in the difference-in-means estimator. In the extreme case, the bias-variance tradeoff may favor a unit-randomized design over a cluster-randomized design. We note that this problem is not unique to $\widehat{\tau}_{DIM}$; the IPS estimator also experiences a tradeoff between the variance reduction due to clustering (due to a higher probability of pure exposure) and the variance increase due to heterogeneity between clusters. In practice the variance is typically reduced by choosing a large number of clusters, since the cluster becomes the effective unit of analysis. Of interest is understanding the trade-off incurred by cluster randomization in this setting, and to design techniques to determine, perhaps from historical data, whether cluster randomization should be used for a given experiment.

# F    Proofs

In this appendix, we provide proofs of all the results in this paper. We apply similar proof techniques to bound the bias in a variety of potential outcomes models, and therefore consider a potential outcomes model which generalizes all the models discussed in this paper:

$$Y_i = g_i(Z_i, e_i).$$

We will first prove some useful lemmas that apply to all potential outcomes models of this form; later we will provide results that are specific to each model type.

## F.1    Useful Lemmas

The first two lemmas in this section will be useful for calculating the bias of the difference in means estimator under the linear, Lipschitz, and $\Delta-$neighborhood functions. The proofs for each bias bound

will follow a similar structure: First, we will decompose the bias of the difference-in-means estimator into the bias contribution of each unit under both treatment and control. Next, we will use the given structure of the potential outcome to either compute the bias exactly (in the linear setting) or bound the bias (in the Lipschitz and $\Delta$-neighborhood settings). In each case, we will be left with a bound in terms of the conditional expectation of the exposure $e_i$ given the treatment status $Z_i$. Our third step will be to relate this quantity to the folded graph clustering objective, $\mathcal{H}(\mathcal{C})$.

We provide lemmas for the first and third steps; the second step is unique to each bias calculation.

**Lemma F.1** (Unit-level bias decomposition). *Let the unit-level responses $Y_i$ be functions of the unit's treatment $Z_i$ and the unit-level exposure $e_i$, so that $Y_i = g_i(Z_i, e_i)$. Then the bias of the difference-in-means estimate of the average treatment effect can be written as*

$$\mathbb{E}[\hat{\tau}] - \tau^* = \frac{1}{N} \sum_{i \in [N]} \mathbb{E}[g_i(Z_i, e_i) - g_i(Z_i, Z_i)|Z_i = 1] - \frac{1}{N} \sum_{i \in [N]} \mathbb{E}[g_i(Z_i, e_i) - g_i(Z_i, Z_i)|Z_i = -1]$$

*Proof.* We begin by writing the bias in terms of the response function $g_i(Z_i, e_i)$:

$$\mathbb{E}[\hat{\tau}] - \tau^* = \mathbb{E}\left[\frac{1}{N_T} \sum_{i \in I} Y_i - \frac{1}{N_C} \sum_{i \in \bar{I}} Y_i\right] - \frac{1}{N} \sum_{i \in [N]} (g_i(1,1) - g_i(-1,-1))$$

$$= \frac{1}{N_T} \sum_{i \in [N]} \mathbb{E}\left[\mathbf{1}\{Z_i = 1\}Y_i\right] - \frac{1}{N_C} \sum_{i \in [N]} \mathbb{E}\left[\mathbf{1}\{Z_i = -1\}Y_i\right] - \frac{1}{N} \sum_{i \in [N]} (g_i(1,1) - g_i(-1,-1))$$

$$= \frac{1}{N_T} \sum_{i \in [N]} \mathbb{E}\left[\mathbf{1}\{Z_i = 1\}g_i(Z_i, e_i)\right] - \frac{1}{N_C} \sum_{i \in [N]} \mathbb{E}\left[\mathbf{1}\{Z_i = -1\}g_i(Z_i, e_i)\right]$$

$$- \frac{1}{N} \sum_{i \in [N]} (g_i(1,1) - g_i(-1,-1))$$

We use the law of total probability to rewrite the expectations as conditional expectations:

$$\mathbb{E}[\hat{\tau}] - \tau^* = \frac{1}{N_T} \sum_{i \in [N]} \mathbb{E}\left[g_i(Z_i, e_i)|Z_i = 1\right] \mathbb{P}(Z_i = 1)$$

$$- \frac{1}{N_C} \sum_{i \in [N]} \mathbb{E}\left[g_i(Z_i, e_i)|Z_i = -1\right] \mathbb{P}(Z_i = -1)$$

$$- \frac{1}{N} \sum_{i \in [N]} (g_i(1,1) - g_i(-1,-1))$$

$$= \frac{1}{N_T} \sum_{i \in [N]} \frac{N_T}{N} \mathbb{E}\left[g_i(Z_i, e_i)|Z_i = 1\right]$$

$$- \frac{1}{N_C} \sum_{i \in [N]} \frac{N_C}{N} \mathbb{E}\left[g_i(Z_i, e_i)|Z_i = -1\right]$$

$$- \frac{1}{N} \sum_{i \in [N]} (g_i(1,1) - g_i(-1,-1))$$

$$= \frac{1}{N} \sum_{i \in [N]} \mathbb{E}\left[g_i(Z_i, e_i)|Z_i = 1\right]$$

$$- \frac{1}{N} \sum_{i \in [N]} \mathbb{E}\left[g_i(Z_i, e_i)|Z_i = -1\right]$$

$$- \frac{1}{N} \sum_{i \in [N]} (g_i(1,1) - g_i(-1,-1))$$

All of the summations are now computing averages over the $N$ terms. We can distribute the terms of the final summation between the first and second summations, which lets us decompose the bias into

contributions from each unit under its control and treated assignments:

$$\mathbb{E}[\hat{\tau}] - \tau^* = \frac{1}{N} \sum_{i \in [N]} \mathbb{E}\left[g_i(Z_i, e_i) - g_i(1, 1) | Z_i = 1\right] - \frac{1}{N} \sum_{i \in [N]} \mathbb{E}\left[g_i(Z_i, e_i) - g_i(-1, -1) | Z_i = -1\right]$$

$$= \frac{1}{N} \sum_{i \in [N]} \mathbb{E}\left[g_i(Z_i, e_i) - g_i(Z_i, Z_i) | Z_i = 1\right] - \frac{1}{N} \sum_{i \in [N]} \mathbb{E}\left[g_i(Z_i, e_i) - g_i(Z_i, Z_i) | Z_i = -1\right]$$

as desired. $\qquad\qquad\square$

**Lemma F.2** (Writing the weighted condition gaps in terms of the graph structure). *Let $e_i$ be the exposure of unit $i$ to treatment, as defined in Eqn (2). Let $\gamma_i \in \mathbb{R}$ be arbitrary. If the treatment assignment vector $\mathbf{Z} \sim \mathcal{D}(\mathcal{C})$ is drawn according to a balanced cluster randomized design (definition 3.1), then the $\gamma_i$-weighted average conditional gap between $e_i$ and the unit's exposure $Z_i$ can be written in terms of the underlying graph weights $w_{is}$ between units assigned to different clusters:*

$$\frac{1}{N} \sum_{i \in [N]} \gamma_i \left(\mathbb{E}\left[Z_i - e_i | Z_i = 1\right] + \mathbb{E}\left[e_i - Z_i | Z_i = -1\right]\right) = \frac{2}{N} \frac{K}{K-1} \sum_{i \in [N]} \sum_{j \notin \mathcal{C}(i)} \gamma_i \sum_s \frac{w_{is}}{\sum_s w_{is}} \frac{w_{js}}{\sum_k w_{ks}}.$$

*Proof.* We begin by observing the useful fact that

$$\sum_s \frac{w_{is}}{\sum_s w_{is}} \sum_j \frac{w_{js}}{\sum_j w_{js}} = 1.$$

Combining this fact with the definition of $e_i$ lets us write all of the terms in this expression as linear combinations of $Z_j$ and $Z_i$, where $Z_i$ is fixed in each conditional expectation.

$$\frac{1}{N} \sum_{i \in [N]} \gamma_i \left(\mathbb{E}\left[Z_i - e_i | Z_i = 1\right] + \mathbb{E}\left[e_i - Z_i | Z_i = -1\right]\right)$$

$$= \frac{1}{N} \sum_{i \in [N]} \gamma_i \left(\mathbb{E}\left[\sum_s \frac{w_{is}}{\sum_s w_{is}} \sum_j \frac{w_{js}}{\sum_j w_{js}} (Z_i - Z_j) | Z_i = 1\right]\right.$$

$$\left. + \mathbb{E}\left[\sum_s \frac{w_{is}}{\sum_s w_{is}} \sum_j \frac{w_{js}}{\sum_j w_{js}} (Z_j - Z_i) | Z_i = -1\right]\right)$$

$$= \frac{1}{N} \sum_{i \in [N]} \gamma_i \sum_s \frac{w_{is}}{\sum_s w_{is}} \sum_j \frac{w_{js}}{\sum_j w_{js}} \left(\mathbb{E}\left[(Z_i - Z_j) | Z_i = 1\right] + \mathbb{E}\left[(Z_j - Z_i) | Z_i = -1\right]\right)$$

$$= \frac{1}{N} \sum_{i \in [N]} \gamma_i \sum_s \frac{w_{is}}{\sum_s w_{is}} \sum_j \frac{w_{js}}{\sum_j w_{js}} \left(2\mathbb{P}\left(Z_j = -1 | Z_i = 1\right) + 2\mathbb{P}\left(Z_j = 1 | Z_i = -1\right)\right)$$

where in the last step we used the fact that $Z_j$ only takes values in $\{-1, 1\}$.

Recall that the $Z_i$ were assigned according to a *cluster-randomized* design $\mathcal{C}$. If $j \in \mathcal{C}(i)$ then $Z_i = Z_j$, so that units in the same cluster contribute zero to the summation above. Otherwise, if $j \notin \mathcal{C}(i)$, we can compute the probability that $Z_i \neq Z_j$. If there are $K$ clusters and $K_T$ of them are chosen to be treated, then we have

$$\mathbb{P}(Z_i = -1 | Z_j = 1 \cap j \notin \mathcal{C}(i)) = \frac{K_C}{K-1}$$

and

$$\mathbb{P}(Z_i = 1 | Z_j = -1 \cap j \notin \mathcal{C}(i)) = \frac{K_T}{K-1}.$$

We can use this information to compute the probabilities in the expression above, applying the fact that $\mathbb{P}(Z_i \neq Z_j | j \in \mathcal{C}(i)) = 0$ to restrict the sum over $j$ to only the units outside of $\mathcal{C}(i)$.

$$\frac{1}{N} \sum_{i \in [N]} \gamma_i \left( \mathbb{E}\left[ Z_i - e_i | Z_i = 1 \right] + \mathbb{E}\left[ e_i - Z_i | Z_i = -1 \right] \right)$$

$$= \frac{2}{N} \sum_{i \in [N]} \gamma_i \sum_s \frac{w_{is}}{\sum_s w_{is}} \sum_{j \notin \mathcal{C}(i)} \frac{w_{js}}{\sum_j w_{js}} \Big( \mathbb{P}(Z_j = -1 | Z_i = 1 \cap j \notin \mathcal{C}(i))$$

$$+ \mathbb{P}(Z_j = 1 | Z_i = -1 \cap j \notin \mathcal{C}(i)) \Big)$$

$$= \frac{2}{N} \sum_{i \in [N]} \gamma_i \sum_s \frac{w_{is}}{\sum_s w_{is}} \sum_{j \notin \mathcal{C}(i)} \frac{w_{js}}{\sum_j w_{js}} \left( \frac{K_C}{K-1} + \frac{K_T}{K-1} \right)$$

$$= \frac{2}{N} \sum_{i \in [N]} \gamma_i \sum_s \frac{w_{is}}{\sum_s w_{is}} \sum_{j \notin \mathcal{C}(i)} \frac{w_{js}}{\sum_j w_{js}} \frac{K}{K-1},$$

as desired. $\qquad\square$

The final lemma in this section writes the exposure vector **e** is a linear combination of the treatment assignments **Z**. This linearity is useful for establishing the connection between the covariance objective and the folded graph objective in Lemma 3.3.

**Lemma F.3.** *Let* **e** *be defined as in Equation* (2). *Then* $\mathbf{e} = C\mathbf{Z}$, *where*

$$C_{ij} = \sum_s \frac{w_{is}}{\sum_s w_{is}} \frac{w_{js}}{\sum_k w_{ks}}$$

*Proof.* The proof proceeds by the definition of $e_i$. Let matrix $B \in [0,1]^{N \times M}$ be defined as

$$B_{is} = \frac{w_{is}}{\sum_s w_{is}}.$$

Then we can write the exposures $e_i$ as a linear combination of the doses $d_s$:

$$e_i = \frac{\sum_s w_{is} d_s}{\sum_s w_{is}}$$

$$= [B\mathbf{d}]_i.$$

Similarly, let matrix $A \in [0,1]^{M \times N}$ be defined as

$$A_{si} = \frac{w_{is}}{\sum_i w_{is}},$$

so that we can write the doses $d_s$ as a linear combination of the treatment effects **Z**:

$$d_s = \frac{\sum_i w_{is} Z_i}{\sum_i w_{is}}$$

$$= [A\mathbf{Z}]_s.$$

Putting these together, we have

$$\mathbf{e} = BA\mathbf{Z}$$

$$=: C\mathbf{Z}$$

with $C_{ij}$ as given in the lemma statement. $\qquad\square$

With these helper lemmas established, we turn to proving the main results in the paper.

## F.2 Proof of Lemma 3.2

We begin by applying Lemma F.1 to decompose the bias in terms of its unit-level contributions:

$$\mathbb{E}[\hat{\tau}] - \tau^* = \frac{1}{N} \sum_{i \in [N]} \mathbb{E}\left[g_i(Z_i, e_i) - g(Z_i, Z_i) | Z_i = 1\right] - \frac{1}{N} \sum_{i \in [N]} \mathbb{E}\left[g_i(Z_i, e_i) - g(Z_i, Z_i) | Z_i = -1\right]$$

We apply the linear response function to simplify this expression:

$$\mathbb{E}[\hat{\tau}] - \tau^* = \frac{1}{N} \sum_{i \in [N]} \mathbb{E}\left[(\alpha_i + \beta_i Z_i + \gamma_i e_i) - (\alpha_i + \beta_i Z_i + \gamma_i Z_i) | Z_i = 1\right]$$

$$- \frac{1}{N} \sum_{i \in [N]} \mathbb{E}\left[(\alpha_i + \beta_i Z_i + \gamma_i e_i) - (\alpha_i + \beta_i Z_i + \gamma_i Z_i) | Z_i = -1\right]$$

$$= \frac{1}{N} \sum_{i \in [N]} \mathbb{E}\left[\gamma_i e_i - \gamma_i Z_i | Z_i = 1\right] - \frac{1}{N} \sum_{i \in [N]} \mathbb{E}\left[\gamma_i e_i - \gamma_i Z_i | Z_i = -1\right]$$

$$= -\frac{1}{N} \sum_{i \in [N]} \gamma_i \left(\mathbb{E}\left[Z_i - e_i | Z_i = 1\right] + \mathbb{E}\left[e_i - (-1) | Z_i = -1\right]\right)$$

Next, we apply Lemma F.2 to write this expression in terms of the graph structure, completing the proof of the first statement of the lemma:

$$\mathbb{E}[\hat{\tau}] - \tau^* = -\frac{2}{N} \frac{K}{K-1} \sum_{i \in [N]} \sum_{j \notin \mathcal{C}(i)} \gamma_i \sum_s \frac{w_{is}}{\sum_s w_{is}} \frac{w_{js}}{\sum_k w_{ks}}. \tag{7}$$

Now we will prove the second part of the lemma, that the folded graph objective provides the bias-minimizing clustering in a minimax sense among all balanced cluster-randomized designs when the potential outcome is linear in $Z$ and the exposure $e$, and the interference parameter $\gamma$ satisfies $\gamma_i = O(1)$. In particular, we will prove this claim when the $\gamma_i$ are bounded on a shared interval $[\Gamma_0, \Gamma_1]$ for all $i$.

We begin by finding the maximum bias (over choice of $\gamma$) for a given clustering $\mathcal{C}$. We use the value of the bias from Eqn (7):

$$\arg\min_{\mathcal{C}} \max_{\gamma \in [\Gamma_0, \Gamma_1]} \left|\mathbb{E}_{\mathbf{Z} \sim \mathcal{D}(\mathcal{C})}[\hat{\tau}] - \tau^*\right|$$

$$= \arg\min_{\mathcal{C}} \max_{\gamma \in [\Gamma_0, \Gamma_1]^N} 2\frac{1}{N} \cdot \frac{K}{K-1} \left|\sum_{i \in [N]} \sum_{j \notin \mathcal{C}(i)} \gamma_i \sum_s \frac{w_{is}}{\sum_s w_{is}} \frac{w_{js}}{\sum_k w_{ks}}\right|.$$

Recall that the edge weights $w_{is}$ are all nonnegative, so term $i$ of the summation takes on the sign of $\gamma_i$. For this reason, the maximum bias occurs when all $\gamma_i$ are of the same sign (so that no terms in the summation cancel each other), at $\gamma_i = \max(|\Gamma_0|, |\Gamma_1|)$.

$$\arg\min_{\mathcal{C}} \max_{\gamma \in [\Gamma_0, \Gamma_1]} \left|\mathbb{E}_{\mathbf{Z} \sim \mathcal{D}(\mathcal{C})}[\hat{\tau}] - \tau^*\right|$$

$$= \arg\min_{\mathcal{C}} \max(|\Gamma_0|, |\Gamma_1|) \cdot 2\frac{1}{N} \cdot \frac{K}{K-1} \sum_{i \in [N]} \sum_{j \notin \mathcal{C}(i)} \sum_s \frac{w_{is}}{\sum_s w_{is}} \frac{w_{js}}{\sum_k w_{ks}}.$$

The terms $K$, $N$, $\Gamma_0$ and $\Gamma_1$ are all constants with respect to the clustering $\mathcal{C}$, so they can be removed without affecting the arg max. We recognize this objective as precisely our folded graph objective

$$\arg\min_{\mathcal{C}} \max_{\gamma \in [\Gamma_0, \Gamma_1]} \left|\mathbb{E}_{\mathbf{Z} \sim \mathcal{D}(\mathcal{C})}[\hat{\tau}] - \tau^*\right| = \arg\min_{\mathcal{C}} H(\mathcal{C})$$

as desired. $\square$

### F.3 Proof of Lemma 4.1 (bounding the bias under the Lipschitz potential outcomes model)

We begin by applying Lemma F.1 to decompose the bias in terms of its unit-level contributions:

$$|\mathbb{E}[\hat{\tau}] - \tau^*| = \left| \frac{1}{N} \sum_{i \in [N]} \mathbb{E}\left[g_i(Z_i, e_i) - g(Z_i, Z_i)|Z_i = 1\right] - \frac{1}{N} \sum_{i \in [N]} \mathbb{E}\left[g_i(Z_i, e_i) - g(Z_i, Z_i)|Z_i = -1\right] \right|$$

$$= \left| \frac{1}{N} \sum_{i \in [N]} \mathbb{E}\left[g_i(1, e_i) - g(1, Z_i)|Z_i = 1\right] - \frac{1}{N} \sum_{i \in [N]} \mathbb{E}\left[g_i(-1, e_i) - g(-1, Z_i)|Z_i = -1\right] \right|$$
$$(8)$$

Next, we use the fact that $g_i(Z, e)$ is $L$-Lipschitz in $e$ to simplify this expression:

$$|\mathbb{E}[\hat{\tau}] - \tau^*| \leq \frac{1}{N} \sum_{i \in [N]} \mathbb{E}\left[L\,|Z_i - e_i|\,\big|\,Z_i = 1\right] + \frac{1}{N} \sum_{i \in [N]} \mathbb{E}\left[L\,|e_i - Z_i|\,\big|\,Z_i = -1\right]$$

$$= \frac{1}{N} \sum_{i \in [N]} L\mathbb{E}\left[|1 - e_i|\,\big|\,Z_i = 1\right] + \frac{1}{N} \sum_{i \in [N]} L\mathbb{E}\left[|e_i - (-1)|\,\big|\,Z_i = -1\right].$$

Observe that $e_i \in [-1, 1]$, so that the terms in absolute values are all positive. This lets us drop the absolute value signs and write

$$|\mathbb{E}[\hat{\tau}] - \tau^*| \leq \frac{1}{N} \sum_{i \in [N]} L\mathbb{E}\left[1 - e_i|Z_i = 1\right] + \frac{1}{N} \sum_{i \in [N]} L\mathbb{E}\left[e_i - (-1)|Z_i = -1\right].$$

We apply Lemma F.2 with $\gamma_i = L$ to bound the bias as

$$|\mathbb{E}[\hat{\tau}] - \tau^*| \leq \frac{2}{N} \frac{K}{K-1} L \sum_{i \in [N]} \sum_{j \notin \mathcal{C}(i)} \frac{1}{\sum_s w_{is}} w_i^T \tilde{w}_j.$$

To prove the second part of the lemma statement (minimax optimality), we compute the maximum bias over $L$-Lipschitz interference functions $g_i$, using the expression for bias given in Equation (8).

$$\arg\min_{\mathcal{C}} \max_{\{g_i \in \text{Lip}_L(e)\}} |\mathbb{E}[\hat{\tau}] - \tau^*| = \arg\min_{\mathcal{C}} \max_{\{g_i \in \text{Lip}_L(e)\}} \left| \frac{1}{N} \sum_{i \in [N]} \mathbb{E}\left[g_i(1, 1) - g_i(1, e_i)|Z_i = 1\right] \right.$$

$$\left. + \frac{1}{N} \sum_{i \in [N]} \mathbb{E}\left[g_i(-1, e_i) - g_i(-1, -1)|Z_i = -1\right] \right|.$$
$$(9)$$

Next, we will show that setting $g_i(Z, e) = L \cdot e \ \forall i$ achieves the maximum over all $L$-Lipschitz functions by computing an upper bound on the argument of the maximum and showing that this choice of $\{g_i\}$ attains that bound. We begin by upper bounding the maximum:

$$\max_{\{g_i \in \text{Lip}_L(e)\}} \left| \frac{1}{N} \sum_{i \in [N]} \mathbb{E}\left[g_i(1, 1) - g_i(1, e_i)|Z_i = 1\right] + \frac{1}{N} \sum_{i \in [N]} \mathbb{E}\left[g_i(-1, e_i) - g_i(-1, -1)|Z_i = -1\right] \right|$$

$$\leq \max_{\{g_i \in \text{Lip}_L(e)\}} \left( \frac{1}{N} \sum_{i \in [N]} \mathbb{E}\left[|g_i(1, 1) - g_i(1, e_i)|\,\big|\,Z_i = 1\right] \right.$$

$$\left. + \frac{1}{N} \sum_{i \in [N]} \mathbb{E}\left[|g_i(-1, e_i) - g_i(-1, -1)|\,\big|\,Z_i = -1\right] \right)$$

$$\leq \frac{1}{N} \sum_{i \in [N]} \mathbb{E}\left[\max_{g_i \in \text{Lip}_L(e)} |g_i(1, 1) - g_i(1, e_i)|\,\big|\,Z_i = 1\right]$$

$$+ \frac{1}{N} \sum_{i \in [N]} \mathbb{E}\left[\max_{g_i \in \text{Lip}_L(e)} |g_i(-1, e_i) - g_i(-1, -1)|\,\big|\,Z_i = -1\right]$$

Next we apply the Lipschitz assumption, the fact that $e_i \in [-1, 1]$, and Lemma F.2 with $\gamma_i = L$:

$$\max_{\{g_i \in \mathrm{Lip}_L(e)\}} \left| \frac{1}{N} \sum_{i \in [N]} \mathbb{E}\left[g_i(1,1) - g_i(1, e_i) | Z_i = 1\right] + \frac{1}{N} \sum_{i \in [N]} \mathbb{E}\left[g_i(-1, e_i) - g_i(-1, -1) | Z_i = -1\right] \right|$$

$$\leq \frac{1}{N} \sum_{i \in [N]} \mathbb{E}\left[L\left|1 - e_i\right| \big| Z_i = 1\right] + \frac{1}{N} \sum_{i \in [N]} \mathbb{E}\left[L\left|e_i - (-1)\right| \big| Z_i = -1\right]$$

$$= \frac{1}{N} \sum_{i \in [N]} L\mathbb{E}\left[1 - e_i \big| Z_i = 1\right] + \frac{1}{N} \sum_{i \in [N]} L\mathbb{E}\left[e_i - (-1) \big| Z_i = -1\right]$$

$$= \frac{2}{N} \frac{K}{K-1} L \sum_{i \in [N]} \sum_{j \notin \mathcal{C}(i)} \sum_s \frac{w_{is}}{\sum_s w_{is}} \frac{w_{js}}{\sum_k w_{ks}}.$$

We have now shown an upper bound on the argument of the right hand side of Equation (9). Next, we will show that $\tilde{g}_i(Z, e) := L \cdot e \ \forall i$ achieves this bound (and is therefore a maximizer over the class of $L$-Lipschitz functions) by directly computing the argument from the right hand side of Equation (9) under this choice of $g$. We use the definition of $\tilde{g}$ to bound the difference in $\tilde{g}$ by the difference in $e$, the fact that $e_i \in [-1, 1]$ to remove the absolute values, and finally apply Lemma F.2 with $\gamma_i = L$.

$$\left| \frac{1}{N} \sum_{i \in [N]} \mathbb{E}[\tilde{g}_i(1,1) - \tilde{g}_i(1, e_i) | Z_i = 1] + \frac{1}{N} \sum_{i \in [N]} \mathbb{E}\left[\tilde{g}_i(-1, e_i) - \tilde{g}_i(-1, -1) | Z_i = -1\right] \right|$$

$$= \left| \frac{1}{N} \sum_{i \in [N]} \mathbb{E}\left[L(1 - e_i) | Z_i = 1\right] + \frac{1}{N} \sum_{i \in [N]} \mathbb{E}\left[L(e_i - (-1)) | Z_i = -1\right] \right|$$

$$= \frac{1}{N} \sum_{i \in [N]} L\mathbb{E}\left[(1 - e_i) | Z_i = 1\right] + \frac{1}{N} \sum_{i \in [N]} L\mathbb{E}\left[(e_i - (-1)) | Z_i = -1\right]$$

$$= \frac{2}{N} \frac{K}{K-1} L \sum_{i \in [N]} \sum_{j \notin \mathcal{C}(i)} \sum_s \frac{w_{is}}{\sum_s w_{is}} \frac{w_{js}}{\sum_k w_{ks}}. \tag{10}$$

We conclude that $g_i(e) = L \cdot e \ \forall i$ achieves the maximum over $L$-Lipschitz functions $g$ in Equation (9). Returning to that statement, we are now able to bound the bias using Equation (10) as

$$\arg\min_{\mathcal{C}} \max_{\{g_i \in \mathrm{Lip}_L(e)\}} |\mathbb{E}[\hat{\tau}] - \tau^*| = \arg\min_{\mathcal{C}} \frac{2}{N} \frac{K}{K-1} L \sum_{i \in [N]} \sum_{j \notin \mathcal{C}(i)} \sum_s \frac{w_{is}}{\sum_s w_{is}} \frac{w_{js}}{\sum_k w_{ks}}.$$

Since $K$, $N$ and $L$ are constants with respect to the clustering $\mathcal{C}$, we recover the minimax optimality of the folded graph clustering:

$$\arg\min_{\mathcal{C}} \max_{\{g_i \in \mathrm{Lip}_L(e)\}} |\mathbb{E}[\hat{\tau}] - \tau^*| = \arg\min_{\mathcal{C}} H(\mathcal{C})$$

### F.4 Proof of Lemma 4.2 (bounding the bias under the $\Delta$−neighborhood potential outcomes model)

We begin by applying Lemma F.1 to decompose the bias in terms of its unit-level contributions:

$$\mathbb{E}[\hat{\tau}] - \tau^* = \frac{1}{N} \sum_{i \in [N]} \mathbb{E}\left[g_i(Z_i, e_i) - g(Z_i, Z_i) | Z_i = 1\right] - \frac{1}{N} \sum_{i \in [N]} \mathbb{E}\left[g_i(Z_i, e_i) - g(Z_i, Z_i) | Z_i = -1\right].$$

Next, we apply the property of the potential outcomes given in (4). We know from the assumption that whenever $|Z - e| < \Delta$, we have $|g_i(Z, e) - g_i(Z, Z)| = 0$, and otherwise we have $|g_i(Z, e) - g_i(Z, Z)| \leq B$. We can therefore bound the absolute value of the bias by $m$ times the probability

that each unit's exposure $e_i$ deviates by more than $\Delta$ from its assignment $Z_i$.

$$|\mathbb{E}[\hat{\tau}] - \tau^*| \leq \frac{1}{N} \sum_{i \in [N]} \mathbb{E}\left[B\mathbf{1}\{e_i < 1 - \Delta\}|Z_i = 1\right] - \frac{1}{N} \sum_{i \in [N]} \mathbb{E}\left[B\mathbf{1}\{e_i > -1 + \Delta\}|Z_i = -1\right]$$

$$= \frac{1}{N} \sum_{i \in [N]} B\mathbb{P}\left(e_i < 1 - \Delta|Z_i = 1\right) - \frac{1}{N} \sum_{i \in [N]} B\mathbb{P}\left(\mathbf{1}\{e_i > -1 + \Delta\}|Z_i = -1\right)$$

$$= \frac{1}{N} \sum_{i \in [N]} B\mathbb{P}\left(1 - e_i > \Delta|Z_i = 1\right) - \frac{1}{N} \sum_{i \in [N]} B\mathbb{P}\left(1 + e_i > \Delta|Z_i = -1\right)$$

We apply Markov's inequality to both terms:

$$|\mathbb{E}[\hat{\tau}] - \tau^*| \leq \sum_{i \in [N]} \frac{B}{\Delta}\left(\mathbb{E}[1 - e_i|Z_i = 1] + \mathbb{E}[1 + e_i|Z_i = -1]\right)$$

Next, we apply Lemma F.2 with $\gamma_i = 1$ to write this expression in terms of our folded graph clustering objective, which completes the proof.

$$|\mathbb{E}[\hat{\tau}] - \tau^*| \leq \frac{2B}{N\Delta} \frac{K}{K-1} \sum_{i \in [N]} \sum_{j \notin \mathcal{C}(i)} \sum_s \frac{w_{is}}{\sum_s w_{is}} \frac{w_{js}}{\sum_k w_{ks}}.$$

$\square$

## F.5 Proof of Lemma 3.3 (The objective $\mathcal{H}(\mathcal{C})$ maximizes the covariance between exposure and treatment assignment)

In this proof, we will use the linearity of $\mathbf{e}$ in $\mathbf{Z}$ to write the covariance objective entirely in terms of linear combinations of $Z_i$. We will then use the fact that $\mathcal{D}(\mathcal{C})$ is a cluster-randomized design to compute the covariance exactly, and show that minimizing the covariance objective is identical to minimizing the folded graph clustering objective.

We begin by rewriting our optimization objective in terms of only the treatment assignments $Z$. Recall that, under the linear dose and exposure mappings, Lemma F.3, $\mathbf{e} = C\mathbf{Z}$ for a known matrix $C$ that depends only on the interference graph.

$$\arg\max_{\mathcal{C}} Tr\left(Cov_{Z \sim \mathcal{D}(\mathcal{C})}(\mathbf{Z}, \mathbf{e})\right) = \arg\max_{\mathcal{C}} Tr\left(\mathbb{E}_{\mathbf{Z} \sim \mathcal{D}(\mathcal{C})}\left[(\mathbf{Z} - \mathbb{E}[\mathbf{Z}])(\mathbf{e} - \mathbb{E}[\mathbf{e}])^T\right]\right)$$

$$= \arg\max_{\mathcal{C}} Tr\left(\mathbb{E}_{\mathbf{Z} \sim \mathcal{D}(\mathcal{C})}\left[(\mathbf{Z} - \mathbb{E}[\mathbf{Z}])(C\mathbf{Z} - \mathbb{E}[C\mathbf{Z}])^T\right]\right)$$

$$= \arg\max_{\mathcal{C}} \mathbb{E}_{\mathbf{Z} \sim \mathcal{D}(\mathcal{C})}\left[Tr\left(C(\mathbf{Z} - \mathbb{E}[\mathbf{Z}])(\mathbf{Z} - \mathbb{E}[\mathbf{Z}])^T\right)\right]$$

Observe that we are taking the trace of a product of two $N \times N$ matrices, $C$ and $(\mathbf{Z}-\mathbb{E}[\mathbf{Z}])(\mathbf{Z}-\mathbb{E}[\mathbf{Z}])^T$. The trace of a product of square matrices is the sum of the entries in their elementwise (Hadamard) product, which lets us write the trace as a sum and apply linearity of expectation:

$$\arg\max_{\mathcal{C}} Tr\left(Cov_{\mathbf{Z} \sim \mathcal{D}(\mathcal{C})}(\mathbf{e}, \mathbf{Z})\right) = \arg\max_{\mathcal{C}} \mathbb{E}_{\mathbf{Z} \sim \mathcal{D}(\mathcal{C})}\left[\sum_{i,j \in [N]} (Z_i - \mathbb{E}[Z_i])(Z_j - \mathbb{E}[Z_j])C_{ij}\right]$$

$$= \arg\max_{\mathcal{C}} \sum_{i,j \in [N]} C_{ij}\mathbb{E}_{\mathbf{Z} \sim \mathcal{D}(\mathcal{C})}\left[(Z_i - \mathbb{E}[Z_i])(Z_j - \mathbb{E}[Z_j])\right]$$

$$(11)$$

We have by Definition 3.1 that $\mathcal{D}(\mathcal{C})$ is a uniform assignment of $K$ balanced clusters into $K_T$ treated units and $K_C = K - K_T$ control units. We see that the value of the expectation $\mathbb{E}_{\mathbf{Z} \sim \mathcal{D}(\mathcal{C})}\left[(Z_i - \mathbb{E}[Z_i])(Z_j - \mathbb{E}[Z_j])\right]$ depends on whether units $i$ and $j$ belong to the same or different clusters. We have

$$\arg\max_{\mathcal{C}} Tr\left(Cov_{Z \sim \mathcal{D}(\mathcal{C})}(\mathbf{e}, \mathbf{Z})\right) =$$

$$\arg\max_{\mathcal{C}} \sum_i \Big( \sum_{j \in \mathcal{C}(i)} C_{ij}\mathbb{E}_{\mathbf{Z} \sim \mathcal{D}(\mathcal{C})}\left[(Z_i - \mathbb{E}[Z_i])(Z_j - \mathbb{E}[Z_j])|j \in \mathcal{C}(i)\right]$$

$$+ \sum_{j \notin \mathcal{C}(i)} C_{ij}\mathbb{E}_{Z \sim \mathcal{D}}\left[(Z_i - \mathbb{E}[Z_i])(Z_j - \mathbb{E}[Z_j])|j \notin \mathcal{C}(i)\right] \Big)$$

Our next step will be to compute both conditional expectations. We will see that the conditional expectations are independent of the indices $i$ and $j$ (since the argument of the expectation depends only on whether $i$ and $j$ belong to the same cluster), and that the conditional expectation is greater when $i$ and $j$ belong to different clusters. This will let us draw an equivalence between our covariance objective and the objective of minimizing the folded graph cut, which will turn out to be exactly the objective of minimizing cuts in $C_{ij}$. We expand the conditional covariances and use linearity of expectation, along with the fact that $\mathbb{E}[Z_j] = (K_T - K_C)/K$ to write

$$
\mathbb{E}_{\mathbf{Z} \sim \mathcal{D}(\mathcal{C})} \left[ (Z_i - \mathbb{E}[Z_i])(Z_j - \mathbb{E}[Z_j]) | j \in \mathcal{C}(i) \right]
$$

$$
= \mathbb{E}_{\mathbf{Z} \sim \mathcal{D}(\mathcal{C})} [Z_i Z_j | j \in \mathcal{C}(i)] - \mathbb{E}_{\mathbf{Z} \sim \mathcal{D}(\mathcal{C})} [Z_i | j \in \mathcal{C}(i)] \, \mathbb{E}[Z_j]
$$

$$
- \mathbb{E}_{Z \sim \mathcal{D}(\mathcal{C})} [Z_j | j \in \mathcal{C}(i)] \, \mathbb{E}[Z_i] + \mathbb{E}[Z_j]\mathbb{E}[Z_i]
$$

$$
= \mathbb{E}_{\mathbf{Z} \sim \mathcal{D}(\mathcal{C})} [Z_i Z_j | j \in \mathcal{C}(i)] - \mathbb{E}[Z_i]\, \mathbb{E}[Z_j] - \mathbb{E}[Z_j]\, \mathbb{E}[Z_i] + \mathbb{E}[Z_j]\mathbb{E}[Z_i]
$$

$$
= \mathbb{E}_{\mathbf{Z} \sim \mathcal{D}(\mathcal{C})} [Z_i Z_j | j \in \mathcal{C}(i)] - \mathbb{E}[Z_j]\mathbb{E}[Z_i]
$$

$$
= \mathbb{E}_{\mathbf{Z} \sim \mathcal{D}(\mathcal{C})} [Z_i Z_j | j \in \mathcal{C}(i)] - \left( \frac{K_T - K_C}{K} \right)^2
$$

$$
= 1 - \left( \frac{K_T - K_C}{K} \right)^2
$$

and similarly

$$
\mathbb{E}_{\mathbf{Z} \sim \mathcal{D}(\mathcal{C})} \left[ (Z_i - \mathbb{E}[Z_i])(Z_j - \mathbb{E}[Z_j]) | j \notin \mathcal{C}(i) \right]
$$

$$
= \mathbb{E}_{\mathbf{Z} \sim \mathcal{D}(\mathcal{C})} [Z_i Z_j | j \notin \mathcal{C}(i)] - \mathbb{E}[Z_j]\mathbb{E}[Z_i]
$$

$$
= 1 \cdot \frac{K_T}{K} \cdot \frac{K_T - 1}{K - 1} + (-1) \cdot \frac{K_T}{K} \frac{K_C - 1}{K - 1} + (-1) \cdot \frac{K_C}{K} \frac{K_T - 1}{K - 1}
$$

$$
+ 1 \cdot \frac{K_C}{K} \frac{K_C - 1}{K - 1} - \mathbb{E}[Z_j]\mathbb{E}[Z_i]
$$

$$
= \frac{K_T(K_T - 1) - K_T(K_C - 1) - K_C(K_T - 1) + K_C(K_C - 1)}{K(K - 1)} - \mathbb{E}[Z_j]\mathbb{E}[Z_i]
$$

$$
= \frac{K_T(K_T - 1) - K_T(K_C - 1) - K_C(K_T - 1) + K_C(K_C - 1)}{K(K - 1)} - \left( \frac{K_T - K_C}{K} \right)^2
$$

$$
= \frac{(K_T - K_C)^2}{K(K - 1)} - \left( \frac{K_T - K_C}{K} \right)^2
$$

Substituting these conditional expectations, we have

$$
\arg\max_{\mathcal{C}} Tr \left( Cov_{\mathbf{Z} \sim \mathcal{D}(\mathcal{C})}(\mathbf{e}, \mathbf{Z}) \right) =
$$

$$
\arg\max_{\mathcal{C}} \sum_i \left( \sum_{j \in [N]} C_{ij} \left( 1 - \left( \frac{K_T - K_C}{K} \right)^2 \right) + \sum_{j \notin \mathcal{C}(i)} C_{ij} \left( \frac{(K_T - K_C)^2}{K(K - 1)} - 1 \right) \right)
$$

Next, we recognize that the sum over all $i$ and $j$ is constant with respect to the cluster randomized design $\mathcal{C}$ and therefore can be removed from the argmax.

$$
\arg\max_{\mathcal{C}} Tr \left( Cov_{\mathbf{Z} \sim \mathcal{D}(\mathcal{C})}(\mathbf{e}, \mathbf{Z}) \right) = \arg\max_{\mathcal{C}} \sum_i \left( \sum_{j \notin \mathcal{C}(i)} C_{ij} \left( \frac{(K_T - K_C)^2}{K(K - 1)} - 1 \right) \right)
$$

$$
= \arg\min_{\mathcal{C}} \sum_i \left( \sum_{j \notin \mathcal{C}(i)} C_{ij} \left( 1 - \frac{(K_T - K_C)^2}{K(K - 1)} \right) \right)
$$

Observe that, as long as there is at least one treated and control unit (i.e., $0 < K_T < K$) then the quantity $1 - \frac{(K_T - K_C)^2}{K(K-1)}$ is positive and can be taken out of the argmax. In this case, we have

$$
\arg\max_{\mathcal{C}} Tr \left( Cov_{\mathbf{Z} \sim \mathcal{D}(\mathcal{C})}(\mathbf{e}, \mathbf{Z}) \right) = \arg\min_{\mathcal{C}} \sum_i \left( \sum_{j \notin \mathcal{C}(i)} C_{ij} \right)
$$

Finally, we substitute the value of $C_{ij}$ from Lemma F.3:

$$\arg\max_{\mathcal{C}} Tr\left(Cov_{\mathbf{Z}\sim\mathcal{D}(\mathcal{C})}(\mathbf{e},\mathbf{Z})\right) = \arg\min_{\mathcal{C}} \sum_i \left(\sum_{j\notin\mathcal{C}(i)} \frac{\sum_s w_{is}\frac{w_{js}}{\sum_k w_{ks}}}{\sum_s w_{is}}\right)$$

which we recognize as precisely the minimum-cut objective on the folded graph. $\square$

## G   Standard deviation and RMSE for experiments

We set up the parameters of our first two experiments so that the error of $\widehat{\tau}_{DIM}$ was almost entirely due to bias, instead of variance. Here we provide the normalized standard deviation and RMSE for the experiments in Sections 5.1 and 5.2, for completeness.

### G.1   Robustness to different graph structures

See Tables 6 and 7.

Table 6: Relative RMSE of $\widehat{\tau}_{DIM}$ as the bipartite stochastic block model changes (see 5.1)

|  | $p = 0.0$ | $p = 0.005$ | $p = 0.05$ | $p = 0.5$ |
|---|---|---|---|---|
| $\mathcal{H}(\mathcal{C})$ | 0.23($\pm$0.03) | 3.84($\pm$0.05) | 11.54($\pm$0.06) | 12.99($\pm$0.09) |
| $\mathrm{Tr}(\mathrm{Var}(\mathbf{d}))$ | 0.28($\pm$0.03) | 3.86($\pm$0.05) | 11.49($\pm$0.06) | 12.92($\pm$0.07) |
| Direct clustering | 0.26($\pm$0.03) | 9.26($\pm$0.14) | 12.69($\pm$0.07) | 12.95($\pm$0.07) |
| EXPOSURE-DESIGN | 0.53($\pm$0.05) | 4.06($\pm$0.07) | 11.9($\pm$0.07) | 13.0($\pm$0.08) |
| Unit-level randomization | 12.43($\pm$0.08) | 12.55($\pm$0.09) | 12.77($\pm$0.07) | 12.96($\pm$0.08) |
| True clusters | 0.31($\pm$0.04) | 3.86($\pm$0.04) | 11.58($\pm$0.06) | 12.96($\pm$0.06) |

Table 7: Relative standard deviation of $\widehat{\tau}_{DIM}$ as the bipartite stochastic block model changes (see 5.1)

|  | $p = 0.0$ | $p = 0.005$ | $p = 0.05$ | $p = 0.5$ |
|---|---|---|---|---|
| $\mathcal{H}(\mathcal{C})$ | 0.23($\pm$0.03) | 0.26($\pm$0.04) | 0.28($\pm$0.04) | 0.42($\pm$0.05) |
| $\mathrm{Tr}(\mathrm{Var}(\mathbf{d}))$ | 0.28($\pm$0.04) | 0.28($\pm$0.03) | 0.31($\pm$0.05) | 0.37($\pm$0.05) |
| Direct clustering | 0.26($\pm$0.03) | 0.69($\pm$0.09) | 0.37($\pm$0.05) | 0.35($\pm$0.05) |
| EXPOSURE-DESIGN | 0.37($\pm$0.04) | 0.33($\pm$0.04) | 0.33($\pm$0.04) | 0.41($\pm$0.05) |
| Unit-level Randomization | 0.39($\pm$0.05) | 0.45($\pm$0.07) | 0.38($\pm$0.05) | 0.41($\pm$0.06) |
| True Clusters | 0.31($\pm$0.03) | 0.23($\pm$0.03) | 0.28($\pm$0.03) | 0.29($\pm$0.04) |

### G.2   Robustness to nonlinearity

See Tables 8 and 9.

## H   Description of the Balanced Partitioning Algorithm

We provide here a brief overview of the clustering algorithms used in our paper. For each of the $\mathcal{H}(\mathcal{C})$, direct clustering, and $\mathrm{Tr}(\mathrm{Var}(\mathbf{d}))$ objectives, we used an implementation of a balanced partitioning algorithm, kindly provided by the authors of Aydin et al. [35], on the appropriately constructed graph.

- For the $\mathcal{H}(\mathcal{C})$ objective, the graph includes only the experimental units as nodes. Each node weight is set to 1, and the edge weight between a pair of nodes is given by Equation (3).

- For the direct clustering, the graph includes both experimental units and interference units as nodes. All experimental units have node weights set to 1, and all interference units have

Table 8: Relative RMSE of $\widehat{\tau}_{DIM}$ as the neighborhood of pure exposure, $\Delta$, widens (see 5.2)

| | $\Delta = 0.1$ | $\Delta = 0.3$ | $\Delta = 0.5$ |
|---|---|---|---|
| $\mathcal{H}(\mathcal{C})$ | 1.0($\pm$0.004) | 0.458($\pm$0.005) | 0.001($\pm$0.0) |
| Tr(Var($\mathbf{d}$)) | 1.002($\pm$0.004) | 0.461($\pm$0.004) | 0.001($\pm$0.0) |
| Direct clustering | 0.997($\pm$0.005) | 0.95($\pm$0.008) | 0.608($\pm$0.02) |
| EXPOSURE-DESIGN | 1.001($\pm$0.004) | 0.509($\pm$0.005) | 0.011($\pm$0.002) |
| Unit-level randomization | 0.998($\pm$0.004) | 1.0($\pm$0.004) | 0.998($\pm$0.003) |
| True clusters | 1.001($\pm$0.004) | 0.459($\pm$0.004) | 0.001($\pm$0.0) |

Table 9: Relative standard deviation of $\widehat{\tau}_{DIM}$ as the neighborhood of pure exposure, $\Delta$, widens (see 5.2)

| | $\Delta = 0.1$ | $\Delta = 0.3$ | $\Delta = 0.5$ |
|---|---|---|---|
| $\mathcal{H}(\mathcal{C})$ | 0.02($\pm$0.003) | 0.024($\pm$0.003) | 0.001($\pm$0.0) |
| Tr(Var($\mathbf{d}$)) | 0.02($\pm$0.002) | 0.021($\pm$0.003) | 0.001($\pm$0.0) |
| Direct clustering | 0.026($\pm$0.003) | 0.038($\pm$0.007) | 0.097($\pm$0.01) |
| EXPOSURE-DESIGN | 0.02($\pm$0.003) | 0.025($\pm$0.003) | 0.005($\pm$0.001) |
| Unit-level randomization | 0.019($\pm$0.002) | 0.019($\pm$0.002) | 0.017($\pm$0.002) |
| True clusters | 0.021($\pm$0.003) | 0.024($\pm$0.003) | 0.001($\pm$0.0) |

node weights set to 0. The edge weight between them is kept as is. The interference units are removed from the clusters once these are computed in order to produce experiment-unit-only clusters.

- For the Tr(Var($\mathbf{d}$)), the graph includes only the experimental units as node. Each node weight is set to 1, and the edge weight between a pair of nodes is given by Equation (5).

In all balanced partitioning runs, the number of clusters is given as fixed, equal to 10 unless specified otherwise. The maximum allowed imbalanced (ratio between the largest and the smallest cluster by sum of node weights) is 10%. The algorithm runs in two steps:

1. An initial embedding of nodes is given onto a line with affinity clustering. This ordering is then broken into clusters by taking contiguous equally-weighted segments of the line.

2. Node swaps are then evaluated to improve the cut size in a post-processing procedure. At most 2 post-processing passes are done before outputting the final clusters.

Please see Aydin et al. [35] for more information. For the EXPOSURE-DESIGN objective, we use the implementation kindly provided by the authors of Harshaw et al. [19]. As described in Section 7.3 of that work, the algorithm uses a greedy local search, starting from singleton clusters, to assign units to clusters. The role of "diversion units" in their paper plays the role of the experimental units in ours, while the "outcome units" in their paper plays the role of the interference units in ours. Their clustering objective in Proposition 7.1 allows for the tuning of a hyper-parameter $\lambda$. We experimented with values $\lambda \in \{0, 0.001, 0.01, 0.1, 1\}$ and reported the best outcome in each case, best defined by minimum mean-squared error unless specified otherwise. To further regularize the output, their implementation adds a regularization term to the objective in the form of $0.95^k \times \sum_C W_C^2$, where $W_C^2$ is the current weight of cluster $C$, and $k$ is the number of iterations completed.