# OpenReview forum: "Cluster Randomized Designs for One-Sided Bipartite Experiments"
_NeurIPS.cc/2022/Conference — NeurIPS 2022 Accept_

### Official Review · Reviewer_pzrY · 2022-07-10

**Rating:** 8
**Confidence:** 3
**Soundness:** 4 excellent
**Presentation:** 4 excellent
**Contribution:** 4 excellent

**Summary:**

The paper studies cluster randomized designs for one-sided bipartite experiments. The authors assume a linear potential outcome model, and a normalized exposure mapping model, and focus on difference-in-means estimators to estimate the average total treatment effect. They propose to find the design that minimizes the bias among balanced K-cluster randomized designs. The authors then demonstrate the robustness of the design to deviations from the assumed potential outcome model/the exposure mapping model. Finally, simulations studies are conducted to show the empirical performance of the design.

**Questions:**

-	I’m curious to see how the proposed design compares to the two-sided designs proposed in [27, 28] (I used the same references as in the paper) Do you believe the proposed design can outperform the two-sided ones? I understand that in practice, it would require different platforms to run one-sided experiments/two-sided ones, so it is not 100% comparable, but I’d love to see some more comparisons and discussions. Thank you!

-	Line 179-184. The authors argue that balancedness helps reduce variance. By how much does balancedness help reduce variance? It will be helpful to see some theoretical/empirical results on this. Or even some more detailed references will be very helpful.

-	Line 238-241. Is the proposed method implemented with the algorithm in [34]? I don’t think this is made 100% clear except a brief mention in line 284, which seems to be referring to how the benchmark methods are implemented. It will be helpful to clarify what specific algorithms are used. Are exact balancedness achieved? If only partial balancedness is achieved, how close is this to exact balancedness?


**Limitations:**

The authors have adequately addressed the limitations and potential negative societal impact of their work.



**Strengths And Weaknesses:**

Strengths

-	The paper is very well-written. The proposed designed is motivated and explained in a very nice way.

-	The paper addresses an interesting and important question: estimating the total treatment effect in the setting of one-sided bipartite experiments.

-	The proposed design appears to have good performance both theoretically and empirically.

-	The paper does a great job discussing robustness to model misspecifications both theoretically and empirically.

Weaknesses

-	I would love to see more detailed discussions on 1. Comparison with two-sided designs. 2. By how much balancedness helps reduce variance. 3. Implementation details (See questions section for more details)

---

> ### Author Response · Authors · 2022-08-02
> **Response to Reviewer pzrY**
>
> We thank the reviewer for their careful reading of our work. We respond to their questions below.
>
> ## Comparison to two-sided experimental design
>
> We thank the reviewer for the question about comparisons to the two-sided randomization designs of [27, 28], in which experimental and interference units are each randomized to treatment and control, but treatment is only applied when a treated unit interacts with another treated unit. The major benefit of this design is that it allows for the quantification of spillover effects by comparing the fully controlled interactions (in which both sides of the interaction were controlled) to the interactions that experience interference (in which only one side of the interaction was assigned to control). In the absence of interference, these interaction types should have the same average response; any deviation from equality indicates interference that can be quantified.
>
> The idea of measuring and correcting for interference is very appealing, especially when interference can be measured using the same experiment that measures the global treatment effect. We think that this line of work is an important part of mitigating interference, and that the two-sided randomization design is a particularly clever way of approaching the problem. One challenge with TSR designs, as discussed by [28, Section 6], is that without further assumptions on the interference model, TSR with treatment fraction p is only able to estimate the global effect of treating p-fraction of the units vs. treating no units, instead of the global effect of treating all units vs. treating no units. [28, Section 8.1] suggests randomizing at the interaction level to allow estimation of the spillover effect at multiple levels of interaction, which could then be extrapolated to estimate the global treatment effect - this of course starts relying on some model for the extrapolation. Ultimately, then, one difference between the TSR design and the cluster-randomized design is that the latter tries to get as many units in “pure treatment” and “pure control” as possible, to estimate the global treatment effect while limiting the need for such extrapolation. As [27, Section 8] suggests, cluster-randomization is likely to outperform TSR when the underlying graph is well-clusterable, but TSR is likely to do better when no cluster-based method can achieve near-complete treatment or control of units.
>
> It would be interesting to think about combinations of TSR and cluster-based designs, particularly in the context of the extrapolation ideas proposed in [28, Section 8.1]. One option would be to cluster-randomize the buyers and sellers individually, in a way that maximizes the variance of the unit-level exposures to treatment under a TSR design (echoing the objective of [18, 19]). The TSR design would allow for the estimation of spillover effects at various levels of interference, while maximizing the variance among the realized levels of interference would improve the accuracy of the extrapolation. We would be happy to mention these interesting research directions at the intersection of the clustering and TSR approaches, and we thank the reviewer for asking this question.
>
> ## How much does balancedness help reduce variance?
>
> Please see our discussion of balancedness in the response to all reviewers.
>
> ## Implementation details
>
> We thank the reviewer for pointing out that our citation of the balanced partitioning algorithm was unclear. We did indeed use the balanced partitioning algorithm of [34], and we would be happy to provide a summary of the algorithm in an appendix for completeness.
>
> The reviewer also asked about the enforcement of the balancedness constraint. The balanced partitioning algorithm of [34] gives strongly balanced clusters. However the Exposure-Design algorithm of [19] only gives weakly balanced clusters, which we addressed by tuning that algorithm’s hyperparameter to encourage balancedness.

---

### Official Review · Reviewer_Vi8h · 2022-07-11

**Rating:** 6
**Confidence:** 4
**Soundness:** 3 good
**Presentation:** 4 excellent
**Contribution:** 2 fair

**Summary:**


In this work, the authors study a new experimental design problem under interference. The underlying graph capturing the interference is a bipartite graph with two types of units (corresponding to the two partitions): experimental and interference units. The goal is to assign the experimental units to treatment or control to estimate the total treatment effect on the experiment units. The only purpose of the interference units is to facilitate interactions between experimental units. The authors motivate this setting with marketplace experiments, where buyers interact with sellers.

For this problem, they study the performance of difference-in-means estimators for cluster-randomized balanced designs, under a suitable linear exposure model. The main claim is a min-max optimal equivalence between treatment effect estimation and identifying a good clustering. In other words, they showed that the clustering that minimizes the maximum bias will result in identifying a partitioning of a weighted graph, with weights representing the strength of interference between the units. They demonstrate the performance of their approaches empirically, along with a robustness study, when the underlying data deviates from the studied exposure model.




**Questions:**

Can the exposure model be tied into some of the earlier literature works? Isn't there double counting with respect to edge weights? (It suggests that the model was chosen after discovering the correct clustering objective)

In Lemma 3.2, are the terms corresponding to gamma0 and gamma1 missing? Although the proof in the appendix states it clearly, I would suggest fixing it to avoid any confusion.

It would be interesting to see the differences in the relative bias (Table 1) with respect to Tr(Var(d)) and H(C) for other synthetic graphs, such as power-law, which exhibit skewed distributions (unlike the stochastic block model used for simulations).



**Limitations:**

As this is a theoretical paper, with simulated experiments, they did not discuss the potential negative societal impact of their work.

**Strengths And Weaknesses:**

The presentation in the paper is excellent, containing extensive discussions on their choice of experimental designs and estimators.

They relate two important quantities of interest to a graph partitioning objective: bias of the estimator and robustness of the design. The robustness is captured using covariance of the exposure vector (under the interference model) and the design vector. From the proofs in the appendix, it is evident that the bias is a constant factor away from the graph partitioning objective, which is computationally challenging to solve exactly, and for which many approximation algorithms are already known in the literature. These new relationships/equivalences are novel and could be helpful in furthering our understanding of experimental designs under interference.

It certainly would be an important addition to strengthen the current work by considering confidence interval bounds for the bias of the considered estimator.

The authors restrict themselves to balanced designs -- it is not immediately clear if this is necessary. They claim that it reduces the variance in the estimator. It would be helpful to understand the exact dependence of parameters here. Moreover, the theoretical claims in the paper do not compare directly with respect to a simple baseline, such as unit-level randomization. As stated, they show the equivalence in terms of the exact clustering, but not the objective values themselves.

Although min-max optimal results are useful, it would be interesting to look at specific cases, for which one can do better than off-the-self graph partitioning (clustering) algorithms from prior literature.

---

> ### Author Response · Authors · 2022-08-02
> **Response to Reviewer Vi8h**
>
> We thank the reviewer for their careful review of our paper, and would like to provide the following responses to the points they raised, copied from their review:
>
> ## Can the exposure model be tied into some of the earlier literature works?
>
> Thank you; we agree with the reviewer that Section 2.2 would benefit from in-line citations to the relevant papers, which are currently only cited in the related works section. The exposure model we consider is related to the linear “dose” function of [18,19] for two-sided bipartite graphs, and also to the model of [9]. We would be happy to update the section to include these citations.
>
> ## Isn't there double counting with respect to edge weights? (It suggests that the model was chosen after discovering the correct clustering objective)
>
> We believe the question is about the clustering objective H(C), which sums over all pairs i, j from distinct clusters, thus counting both the interference of i on j and the interference of j on i. We observe that these two directed interferences may in fact be of different magnitude, so that counting both directions is important to reducing bias. This occurs because the normalization factors may be different for each unit; for example, unit i’s only connection may be to interference unit s, which is one of many neighbors of unit j. In this scenario, $Z_j$ would have a stronger influence on $e_i$ than $Z_i$ would have on $e_j$.
>
> In Lemma 3.2, are the terms corresponding to gamma0 and gamma1 missing? Although the proof in the appendix states it clearly, I would suggest fixing it to avoid any confusion.
> The terms gamma0 and gamma1 can be seen in the first max of the equation after line 223. The key proof idea here is that H(C) is linear in the $\\gamma_i$, so that maximizing over all $\\gamma_i$ simultaneously will set each $\\gamma_i$ to the same constant, which can be removed from the argmax. This is why the gamma0 and gamma1 disappear in the middle and right hand sides of that same equation.
> We would like to ensure that our results are as clear as possible, and it is possible that we misunderstood what the reviewer is asking. Please let us know what we can do to clarify this further, if any questions remain.
>
> ## It would be interesting to see the differences in the relative bias (Table 1) with respect to Tr(Var(d)) and H(C) for other synthetic graphs, such as power-law, which exhibit skewed distributions (unlike the stochastic block model used for simulations).
>
> Thank you for this great suggestion. We have performed a new experiment on a bipartite power-law graph with latent classes, generated by combining the bipartite preferential attachment model in [1] with the “preferential attachment in graphs with affinities” model in [2].
>
> [1] Guillaume, Jean-Loup, and Matthieu Latapy. "Bipartite graphs as models of complex networks." Physica A: Statistical Mechanics and its Applications 371.2 (2006): 795-813.
> [2] Lee, Jay, et al. "Preferential attachment in graphs with affinities." Artificial Intelligence and Statistics. PMLR, 2015.
>
> In our experiment both the Tr(Var(d)) and the H(C) objectives outperformed the unit-level randomization. We have the following results for relative bias (as in our other experiments, variance was inconsequential; standard deviation was 0.1 for all results):
>
> $$
> \\begin{array}{l|r|r}
>     \\text{Design}   &  \\text{Graph with strong latent structure} & \\text{Graph with weak latent structure}\\\\\\hline
>     \\text{H(C)} &  1.9  &  3.5\\\\
> \\text{Tr(Var(d))}  &  2.0 & 3.7 \\\\
> \\text{Unit-level}  &  4.9  & 5.4 \\\\
> \\text{True Clusters} & 2.9  & 5.2
> \\end{array}
> $$
>
> Interestingly, in this setting we see that the cluster-randomized designs outperform the true latent clusters - this is possible because the interference occurs with respect to a single draw of the random graph, which the clustering algorithm gets to see. We speculate that factors specific to the power-law graphs, notably the existence of vertices with very high degree, might cause the optimal clustering for a given draw of the graph to be very different from the optimal clustering for the graph on average. This could have negative consequences for experimental design when only a random draw of the edges is observed, but the interference occurs according to the underlying latent structure.
> We were also surprised that the Tr(Var(d)) objective performed on par with the H(C) objective; we had expected the former to do worse in a setting where the degree distribution was very different between units. It is possible that the small size of the graph we used (100 experimental units, with degrees distributed Zipf(3)) led to an insufficiently broad range of normalizations to distinguish between the two objectives.
> We thank the reviewer for suggesting this experiment, and we would be happy to include it in an appendix if the reviewer thinks it would be helpful.

---

> > ### Comment · Reviewer_Vi8h · 2022-08-08
> > **Rebuttal Response**
> >
> > I thank the authors for answering my questions with great clarity -- I am raising my score.
> >
> > I would suggest the authors to include your additional experiment on power-law graphs and your response to the choice of balanced design, appropriately in the revised version.
> >
> > Yes, I went through the proof of Lemma 3.2 in the appendix. It was clear from the proof about the values of gamma1 and gamma0. However, as stated in the main version, these variables do not seem to have any specific meaning. Is it possible to rephrase it using order notation?

---

> > > ### Author Response · Authors · 2022-08-08
> > > **Thanks; we will make the suggested revisions**
> > >
> > > Thank you for the response. We will include the power-law graph experiments and the discussion of the choice of balanced design in our revision.
> > >
> > > Thank you also for clarifying your comment on the values Gamma0 and Gamma1. We now understand that you were asking for alternative notation to remove the clutter caused by these two dummy variables. We agree with you that this is a good change that will improve the readability of the result, and we will change the lemma to read $\\gamma_i = O(1)$. We will also make the corresponding notational change in the proof.

---

> ### Author Response · Authors · 2022-08-02
> **Continuation of Response to Reviewer Vi8h**
>
> ## It certainly would be an important addition to strengthen the current work by considering confidence interval bounds for the bias of the considered estimator
>
> Since confidence intervals are taken with respect to some source of randomness, while the bias is an expectation, we interpret this question as asking about confidence intervals with respect to some underlying random model on the parameters. For example, if we assumed that the parameters $\\gamma_i$ in the linear model were drawn from N(0,1), then the bias would be a weighted sum of Gaussians, and we could use concentration inequalities on such sums to put confidence intervals on the bias. We would be happy to mention this in the text.
>
> ## The authors restrict themselves to balanced designs -- it is not immediately clear if this is necessary. They claim that it reduces the variance in the estimator. It would be helpful to understand the exact dependence of parameters here.
>
> Please see our discussion of balanced designs in the response to all reviewers.
>
> ## Moreover, the theoretical claims in the paper do not compare directly with respect to a simple baseline, such as unit-level randomization. As stated, they show the equivalence in terms of the exact clustering, but not the objective values themselves.
>
> Thank you for pointing this out. Please see our discussion above in the response to all reviewers.
>
> ## Although min-max optimal results are useful, it would be interesting to look at specific cases, for which one can do better than off-the-shelf graph partitioning (clustering) algorithms from prior literature.
>
> We agree with the reviewer that it is informative to consider how our clustering algorithm compares with prior literature on specific graph instances. We have provided two such examples in Appendix C: C.1 demonstrates a graph on which our clustering algorithm outperforms the naive balanced partitioning that treats the bipartite graph as a non-bipartite graph, while C.2 demonstrates a graph on which our clustering algorithm outperforms the bipartite clustering algorithm of [18, 19] that is designed for two-sided bipartite experiments.

---

### Official Review · Reviewer_nu8Q · 2022-07-11

**Rating:** 6
**Confidence:** 4
**Soundness:** 3 good
**Presentation:** 3 good
**Contribution:** 2 fair

**Summary:**

The authors examine a setting of experimentation they refer to as "one-side bipartite experimentation", in which treatment and outcomes are measured on one set of units, but connections between units (and, therefore, interference) operates through a distinct set of units. The authors provide theoretical results and simulations which demonstrate that clustering on a special objective function can effectively allow for inference in this setting with minimal bias.


**Questions:**

The authors claim "variance is inconsequential in this setting". Surely there are scope conditions or assumptions necessary for this statement to hold. I think the authors may be assuming that methods compared are methods of clustered-RA (or unit-level RA where bias is extremely large)? If all clustering methods use the same number of clusters, this would be true, but I don't see why that would necessarily be the case if one were choosing the number of clusters as a practitioner would be (e.g. through finding an elbow, etc). The appropriate number of clusters could look quite different depending on the clustering objective. In any event, I'd appreciate the authors to clarify this (also in the text).

**Limitations:**

Methods like this which allow for better and more accurate experimentation are a key way to detect and analyze biases in complex real-world systems.

**Strengths And Weaknesses:**

Originality:
- The theoretical analyses are not particularly novel.
- The setting, while interesting, is pretty straightforward: the one-sided bipartite graph just implies a straightforward way to construct an exposure graph. The methods thereafter are fairly straightforward.

Quality:
+ The authors do a good job of drawing out the implications of their main clustering result
+ The focus on design rather than analysis is very welcome.

Clarity:
+ Found the paper quite easy to follow.

Significance:
+ The problem setting is interesting and relevant to many practitioners.

Section 2.1 defines a model which is, essentially, the "Linear-in-Means" model of [1]. This important related work should be discussed. In particular, [1] draws out an agnostic basis for this model. Section 2.2 also has a discussion of exposure maps which greatly parallels section 3.1 of [1]. This is not an accusation of plagiarism, to be clear, but the paper would be improved by drawing more heavily on this.

Greater discussion of the estimand under less favorable data generating processes (in the vein of [2]), could be an interesting direction to make the work more useful in practice. The simple SBM of section 5.2, for example, obviously aligns nicely with a clustering-based design.

[1] A. Chin, Regression adjustments for estimating the global treatment effect in experiments with interference, Journal of Causal Inference, May 2019. https://arxiv.org/abs/1808.08683
[2] Fredrik Sävje. Peter M. Aronow. Michael G. Hudgens. "Average treatment effects in the presence of unknown interference." Ann. Statist. 49 (2) 673 - 701, April 2021. https://doi.org/10.1214/20-AOS1973

---

> ### Author Response · Authors · 2022-08-02
> **Response to Reviewer nu8Q**
>
> We would like to thank the reviewer for their close reading of our work, and offer the following responses to the points they raised.
>
> ## The one-sided bipartite graph implies a straightforward way to construct the exposure graph
>
> Please see our discussion in the combined reviewer response.
>
> ## Relationship to paper [1]
>
> We thank the reviewer for drawing our attention to [1], which we had not seen and which is indeed an important piece of related work that should be included in our paper. One notable difference between our model and the Linear-In-Means model of [1] is that our model allows the parameters $\\alpha_i$, $\\beta_i$, and $\\gamma_i$ to vary among the units i, whereas the model of [1] uses common parameters. This distinction allows [1] to estimate the parameters of the (common) linear model, and use these estimates to extrapolate to the “fully treated” or “fully controlled” setting.
>
> We had considered modeling interference in the same way as [1], using common linear model parameters across all units i. In this setting, it would even be possible to use the principles of optimal linear design to choose a clustering that minimizes the variance (and thus the MSE) of the ordinary least squares estimator. However, we ultimately decided to proceed with our model because it did not rely on any shared response structure across the units. The agnostic perspective of [1] is particularly interesting to us because it provides a rationale for using the LIM model even when we don’t believe the generative model itself is linear - precisely the concern we had about relying on the assumption of common parameters across units. We again thank the reviewer for pointing us to this work; we will both include it in the paper discussion and use it to inform our future work in this area.
>
> ## Discussion of the design under less favorable data generating processes
>
> The reviewer makes a good point about studying our design under less favorable data-generating processes. The field has developed several tools to address interference, and different tools work better in different settings. Cluster randomized designs work best when the underlying graph has some clustering structure, so that it is possible to assign treatments in such a way that some units are near-completely controlled while others are near-completely treated. If that is not possible, then a cluster-randomized design will not provide a meaningful benefit over unit-randomized design, and other tools must be used instead. For example, a stronger model assumption on the potential outcomes lets us estimate the effect of complete treatment or control while only ever observing units with partial exposure. Another alternative is to change the mechanism of randomization itself, such as with two-sided randomized designs and the associated adjusted estimators (see our references [27, 28]). We appreciate the reviewer’s interest in this point, and believe that understanding how to choose between the methods available to address interference, or even combining them, is a useful area of future work.
>
> ## The role of cluster size in the estimator’s variance
>
> Please see our discussion under “Motivation from variance reduction” in the main reviewer response. The reviewer makes a good point about hyperparameter selection to define the number of clusters, and we would be happy to  incorporate this discussion into the text.

---

### Official Review · Reviewer_qE7Y · 2022-07-16

**Rating:** 4
**Confidence:** 2
**Soundness:** 2 fair
**Presentation:** 3 good
**Contribution:** 2 fair

**Summary:**

The paper studies cluster randomized signs for one-sided bipartite experiments under network interferences. The paper discussed many different models, but I will only define the main one studied. There is a weighted bipartite graph between a set of experiments and a set of interference units, with the weight w_{is} between the experiment i and interference unit s indicating relationship between i and s.  For each experiment i, we can assign either "control" (Z_i = -1) or "treat" (Z_i = 1) to i.  The outcome for the i-th experiment is a real number Y_i(Z) = alpha_i + beta_i Z_i  + gamma_i e_i(Y). The alpha_i + beta_i Z_i is a linear function of the treatment Z_i for i, and e_i is called the exposure, which captures the interferences of the (Z_j)_{j\neq i} values to the outcome for i. e_i(Y) is called the exposure mapping.

The paper propose a bipartite analogue of the neighborhood-based exposure mapping. In this model, the dose of an interference unit s is computed as the weighted average of the treatment assignment Z_i among the neighboring experiments i of s, where weights are w_{is} values. Then, the exposure e_i is the weighted average of the doses of neighboring interference units s of i, again using w_{is} values.

The average total treatment effect tau is defined as 1/N * sum_{i \in [N]} (Y_i(vector Z = all 1 vector) - Y_i(vector Z = all -1 vector)). In the difference-in-means (DIM) estimator for tau, we choose randomly N_T experiments to treat, and N_C experiments to control, where N_T and N_C are numbers chosen upfront. The estimation given by the DIM estimator is \hat \tau_DIM := \sum_{i:Z_i = 1} Y_i/ N_T - \sum_{i:Z_i = - 1} Y_i/N_C.  The bias of the estimation is defined as \tau - \hat \tau_DIM.

The main goal of the paper is to use balanced clustering to reduce the mean squared error (MSE) of the DIM estimator. Let C be a partition of the N experiments into K equally sized clusters.  A balanced k-cluster randomized design D(C) is a distribution over vectors Z \in {-1, 1}^N, such that experiments in the same cluster have the same +-1 values, and there are K_T clusters with +1 values. The authors gave the formula for the bias under the cluster design, presented some robustness results, and conducted experiments on the effectiveness of the method.

**Questions:**

It would be good to compare the clustering-based algorithm and the original one (without the clustering used) theoretically.  This could be done using an example, or a theorem that covers typical cases that arise in practice.

**Limitations:**

Not applicable.

**Strengths And Weaknesses:**

Evaluations:  The main contribution of the paper is use of the one-sided cluster design algorithm to reduce the bias of DIM estimator.  However, after giving the formula and the optimization problem in Lemma 3.2, I fail to see why this algorithm is better than the standard one without the clustering. A lemma about some typical cases or even an example suggesting this would be helpful.  Moreover, the clustering idea was not new. It is used in cases where all units are experimental units, and where there are experimental and interference units, but the outcomes are measured in the interference units.  It is not so hard to extend the idea to the setting studied in this paper.

I am also annoyed by the bad format of the supplementary material. There are so many places where the mathematical formulas exceed the width of a line by a lot. The authors may have submitted the paper (at least the supplementary material) in a hurry.

---

> ### Author Response · Authors · 2022-08-02
> **Response to Reviewer qE7Y**
>
> We would like to thank the reviewer for their careful reading of our paper. Our response to the three points raised by the reviewer are as follows:
>
> ## Comparison to unit-level randomization
>
> Please see our discussion in the combined reviewer response.
>
> ## Novelty of the clustering idea
>
> Please see our discussion in the combined reviewer response.
>
> ## Supplementary material formatting
>
> We apologize for the extended line lengths, and we will fix the typesetting in the supplement.

---

> > ### Author Response · Authors · 2022-08-10
> > **Seeking feedback**
> >
> > We thank the reviewer for their comments. It would be great if you can comment on our response addressing the reviewer's major concerns. Thanks.

---

### Author Response · Authors · 2022-08-02
**Response to all reviewers**

We thank the reviewers for their thoughtful and engaged reviews of our work. Each reviewer raised several important points. We will begin by addressing three themes that were shared across several reviewers. Additional responses to individual reviewer comments can be found as direct replies to those reviews.

---

> ### Author Response · Authors · 2022-08-02
> **Motivation for balancedness**
>
> Reviewers nu8Q, Vi8h and pzrY had questions about our restriction to balanced designs, and specifically how balancedness related to the estimator’s variance. The use of balanced clustering designs has several motivations, which we expand upon here.
>
> ## Motivation from Variance Reduction
>
> From the theoretical side, a balanced design provides control of the variance by ensuring that the treated fraction of units is roughly constant across randomizations.  If the clusters are highly imbalanced and a fixed number $K_T$ of clusters are assigned to treatment, then the actual fraction of units assigned to treatment could vary significantly, increasing the variance of the treatment effect estimator. This idea is further formalized in Section 4.2 of [9].
>
> As Reviewer nu8Q correctly points out, balanced clustering only reduces the variance under some assumptions about the interference effects gamma_i. In Appendix D we describe how clustering may actually increase variance if the gamma_i are correlated among clusters. We believe that developing methods to quantify the benefit of balanced clustering for a specific problem instance, or to choose the correct number of clusters, is an important area of future work.
>
> In some experimental settings, pre-experiment data is predictive of the estimator variance. For example, in a setting where the variance of outcomes before the experiment starts is much larger than the anticipated magnitude of effects, pre-experiment data can be used to evaluate the variance characteristics of a given clustering (e.g. using an A/A test). In this way, the practitioner can estimate the effect on the variance of a given clustering for their data set, in order to determine the value of  a balanced clustering over unbalanced, or even no clustering.
>
> ## Implementation Considerations
>
> Practitioners may appreciate two benefits of balanced designs beyond the variance reduction mentioned above. First, when exactly K_T of K clusters are treated, balancing the clusters ensures control over the fraction of units that are treated. Controlling this fraction is important when we want to balance the scientific value of experimentation with potential negative effects on the treated units (i.e. staying within the experimental budget). Secondly, it has been our experience that many clustering algorithms that do not control for balancedness and cardinality sometimes produce many singletons clusters, when clustering incentives are not strong enough to group these units with other units. This is what we observed with the Exposure-Design objective of [18, 19] when its hyperparameters are not tuned properly. When datasets are quite large, the large number of singleton clusters produced by these algorithms can slow down certain data analysis pipelines that work better in low cardinality settings. If these singletons clusters are to be clustered to reduce cardinality without improving any “cut”-like objective, it may make sense to do so in a balanced way for any of the reasons listed above. In other words, the occasional practical need to control for cluster cardinality is extra motivation to maintain balance instead of an arbitrary grouping of isolated nodes
>
> We would be happy to include further discussion around balancedness in the main text or in appendix in an updated version of the paper.

---

> ### Author Response · Authors · 2022-08-02
> **Theoretical comparison to unit-level randomization**
>
> Reviewers qE7Y and Vi8h both asked for a comparison between the biases of the cluster-randomized design and the unit-randomized design. Unit-level randomization can be thought of as a special case of balanced cluster-randomized design in which there are N clusters of one unit each. As a result, the bias of the unit-level randomized design can be derived from Lemma 3.2 by replacing K with N and $j \\not\\in C(i)$ with $j \\neq i$. Letting $X_ij$ denote the influence of $Z_j$ on $e_i$ as given in our Eqn (2), we see that the improvement in bias from clustering is given by the average over N terms of the form:
>
> $\\frac{1}{N} \\sum_{i\\in [N]}\gamma_i  (\\sum_{j\\not\\in C(i)} X_ij (\\frac{N}{N-1} - \\frac{K}{K-1}) + \\sum_{j\\in C(i), j\\neq i} X_ij \frac{N}{N-1})$
>
> Each of the N terms is composed of two terms of opposite signs. The first term captures the fact that, under cluster-level randomization with a fixed number of treated clusters, an element j belonging to a different cluster as unit i is more likely to satisfy $Z_i = Z_j$ when the number of clusters is large. This effect becomes negligible for large experiments, going to 1/N as N goes to infinity if K scales with N. The second term captures the fact that a unit j that belonged to i’s cluster under the cluster randomized design will now contribute to the bias of the unit-randomized design. The cluster-randomized design will have lower bias than the unit-randomized design to the extent that this term is large - i.e. that the interference values $X_ij$ within a single cluster are large on average.
>
> We would be happy to  add this as either a corollary or an appendix to the paper, since multiple reviewers were interested in such a result.

---

> ### Author Response · Authors · 2022-08-02
> **Novelty of the clustering objective**
>
> Reviewers qE7Y and nu8Q inquired to what extent our suggested clustering objective is not a simple extension of existing ideas surrounding cluster-randomized designs. Our paper is certainly not the first to suggest cluster-randomized designs (see our review in lines 74-87). However, there are many possible choices of clustering algorithms, so we believe that the choice of the clustering objective is an important contribution.
>
> In particular, in Section 3.1 we describe the natural extensions of two existing cluster-randomized designs to our bipartite setting, and we provide counterexamples showing that these two objectives fail to minimize the bias. First, the approach of directly clustering the bipartite graph (ignoring the bipartite structure) fails because it considers only one-hop neighbors, but interference in a bipartite graph is fundamentally at least a two-hop phenomenon. Second, an existing cluster randomization objective for the two-sided bipartite design, used in [18, 19] fails in our one-sided design setting because it can myopically focus on only the cluster assignment of the highest-weighted experimental units. See Section 3.1 and Appendix C for further discussion. We believe this illustrates the importance and nontriviality of choosing the correct clustering objective.

---

### Meta-Review · Area_Chair_d7Hz · 2022-08-26

**Recommendation:** Accept
**Confidence:** Less certain

**Metareview:**

This well-written paper proposes a possibly-new experiment-design problem where there is interference. This interference is modeled by a bipartite graph where one side has the "experimental" units and the other has "interference" units. The purpose of the interference units is to facilitate interactions between the experimental units. The goal is to assign the experimental units to "treatment" or "control" in order to estimate the total treatment effect on the experiment units. Specifically, for each experiment i, we can either assign "control" (Z_i = -1) or "treat" (Z_i = 1). The outcome for this experiment is the value Y_i(Z) = alpha_i + beta_i Z_i + gamma_i e_i(Y)---where e_i(Y) is called the "exposure mapping" that captures the interferences of the (Z_j: j distinct from i) values with the outcome for i. This setting is motivated by marketplace experiments where buyers interact with sellers.

This primarily-theoretical paper studies the performance of difference-in-means estimators for cluster-randomized balanced designs, under a "linear exposure" model. One key results is a min-max optimal equivalence between treatment effect estimation and identifying a good clustering: the clustering that minimizes maximum bias will basically yield a partitioning of a weighted graph, with weights representing the strength of interference between the units. Simulation results are also given.

There are concerns about the novelty of the clustering objective, and the supplementary material has poor formatting, but the paper's contributions are appreciated. The authors are asked to carefully incorporate the referee-comments.

**Award:**

No

---

### Decision · Program_Chairs · 2022-09-14

Accept